# Anthropogenic forcing and response yield observed positive trend in Earth's energy imbalance

Shiv Priyam Raghuraman [1✉], David Paynter[2] & V. Ramaswamy [1,2]

The observed trend in Earth's energy imbalance (TEEI), a measure of the acceleration of heat uptake by the planet, is a fundamental indicator of perturbations to climate. Satellite observations (2001–2020) reveal a significant positive globally-averaged TEEI of $0.38 \pm 0.24$ $Wm^{-2}decade^{-1}$, but the contributing drivers have yet to be understood. Using climate model simulations, we show that it is exceptionally unlikely (<1% probability) that this trend can be explained by internal variability. Instead, TEEI is achieved only upon accounting for the increase in anthropogenic radiative forcing and the associated climate response. TEEI is driven by a large decrease in reflected solar radiation and a small increase in emitted infrared radiation. This is because recent changes in forcing and feedbacks are additive in the solar spectrum, while being nearly offset by each other in the infrared. We conclude that the satellite record provides clear evidence of a human-influenced climate system.

[1] Program in Atmospheric and Oceanic Sciences, Princeton University, Princeton, NJ, USA. [2] NOAA Geophysical Fluid Dynamics Laboratory, Princeton, NJ, USA. ✉email: shivr@princeton.edu

Earth's energy imbalance (EEI) is the difference between the incoming solar radiation ($S_0$), and the reflected shortwave radiation (RSW) plus the outgoing longwave radiation (OLR), at the top of the atmosphere[1–3]. Thus, EEI is a fundamental measure of the degree to which the Earth's global climate system is out of balance. A trend in EEI measures the acceleration of heat uptake by the planet and hence is an indicator of perturbations to the coupled atmosphere-ocean-land-ice system[4–8]. The mean EEI during 2005–2015 was estimated to be $0.71 \pm 0.10$ Wm$^{-2}$. Oceans store 90% of this excess heat[9]. Because of this close relationship between EEI and ocean heating, EEI trends have an important bearing on the warming of the global climate system, sea-level rise, and marine health[3]. In addition, understanding historical trends in EEI and its components improves the modeling of the projections of future climate change[10,11], which in turn forms the basis for policymaking and mitigation efforts[12].

In the contemporary climate system, internal variability ($\epsilon$), effective radiative forcing ($\Delta ERF$), and the radiative response ($\lambda \Delta T_s$) change EEI[13–16]. Thus, an anomaly in EEI can be expressed as the sum of these three terms.

$$\Delta EEI = \Delta ERF + \lambda \Delta T_s + \epsilon \qquad (1)$$

Climate feedbacks and surface temperature are represented by $\lambda$ and $\Delta T_s$, respectively. Hereafter, the term radiative forcing refers to the effective radiative forcing, which comprises external forcings including natural (e.g., solar and volcanic) and anthropogenic (e.g., well-mixed greenhouse gases and aerosols), as well as rapid adjustments in response to the forcing[13].

Satellite observations by Clouds and the Earth's Radiant Energy System Energy Balance and Filled (CERES EBAF) have provided an uninterrupted two-decade-long time series (January 2001–December 2020) of EEI along with $S_0$, RSW, and OLR, allowing the study of TEEI[17]. While the mean EEI in CERES data is adjusted to be consistent with in-situ ocean observations, the interannual variations and trends observed by CERES are reliable, as evidenced by the excellent agreement between the three satellites CERES derives its data from refs. [17,18]. While we know that TEEI is influenced by internal variability in the climate system, external forcings, and climate feedbacks, the extent of the contribution from each has not been previously determined[19–30]. In particular, due to the inherent noise in the Earth system, a single observed 20-year time series of EEI is only one of many possible time series that internal variability could produce[31,32], and therefore, it is imperative to quantify the contribution of internal variability ($\epsilon$) to TEEI.

We focus here on the investigation of the trend in EEI (TEEI). In particular, the contributions to TEEI by the drivers of the OLR and RSW trends, are an unexplored and unquantified area in the explanation of the observed satellite record. We use Coupled Model Intercomparison Project Phase 6 (CMIP6) experiments and design a hierarchy of climate model experiments with the Geophysical Fluid Dynamics Laboratory Coupled/Atmospheric Model 4.0 (GFDL CM4/AM4)[11,33] to better understand the contributions of the three components of Eq. (1) to the CERES-observed TEEI, thereby providing an assessment of the relative importance of anthropogenically induced changes versus internal variability changes. We show that anthropogenic forcing and the associated climate response yield the observed positive TEEI.

## Results

### Trends in CERES observations.
The trends in EEI (TEEI), RSW, and OLR are computed by taking the linear fits of the $\Delta EEI$, $\Delta RSW$, and $\Delta OLR$ time series, respectively. The observed $\Delta EEI$ time series yields a significant positive trend of $0.38 \pm 0.24$ Wm$^{-2}$decade$^{-1}$

(Figs. 1a, 2, 3a, 4 and Supplementary Fig. 1a; uncertainty given by 95% confidence interval (CI); hereafter all mentions of uncertainty are given by CI unless otherwise specified). The 95% CI consists of uncertainty due to observational error, as well as uncertainty due to internal variability (latter quantified by standard error associated with linear fit[17,30,34,35]) (see "Methods" section). Although there is excellent agreement between the individual satellites CERES derives its data from, there is, however, the potential for systematic errors associated with the observed trend due to instrument drift. We attach an estimate of 0.20 Wm$^{-2}$decade$^{-1}$ (assuming a normal distribution) to CERES trends, based on best realistic appraisals of observational uncertainty (N. Loeb, CERES Science Project Lead, personal communication)[36–38].

The observed TEEI is nearly 40% of the mean EEI of $1.00 \pm 0.17$ Wm$^{-2}$ over this period. This TEEI is consistent with ocean observations that showed an increase in ocean heat uptake over the last two decades[39,40]. The trends for all latitudes have a positive TEEI, which indicates an increase in radiative energy across the system (Fig. 3 and Supplementary Fig. 2d). Nearly half of the global trend comes from the tropics with the extra-tropics and poles making up the other half of the global trend (Fig. 3a). Radiatively, this trend is driven by large reductions in RSW, compensated only by a relatively weaker increase in OLR (Fig. 1b, c, Fig. 3b, c). Later, we will use AM4 model experiments to understand the drivers of these trends. GFDL CM4/AM4 has been shown to simulate radiation, clouds, and precipitation with relatively small biases amongst CMIP6 models, when compared to observations, (Figs. 18–19 in Boucher et al. 2020[41]). GFDL AM4 simulations of EEI, RSW, and OLR are validated against CERES observations and are in excellent agreement (Supplementary Fig. 3). Furthermore, as will be elaborated upon in the next section, our GFDL CM4/AM4 estimates of internal variability are robust when compared against CMIP6 estimates.

### The observed trend in EEI is unexplained by internal variability.
Our hierarchy of modeling experiments allows us to investigate the possible contributions of trends in $\Delta ERF$, $\lambda \Delta T_s$, and $\epsilon$ to TEEI. We compose five sets of estimates of the internal variability ($\epsilon$) using climate model simulations. For all five sets of estimates, we calculate $\epsilon$ as the $\pm 2\sigma$ range of 20-year trends across the realizations. Two sets of estimates $\epsilon$ come from CMIP6 fully coupled model simulations (which have freely evolving SSTs and sea ice): one with forcing agents at 1850 levels (CMIP6 Control) and historical forcing in another (CMIP6 Historical). These provide estimates of $\epsilon$ in coupled models in the preindustrial era (CM PI) and present-day (CM PD). Three sets of estimates of $\epsilon$ come from atmosphere-only model experiments that we conducted with the GFDL model (AM4 Control, AM4 PSST, AM4 PSST+ERF). Here, we investigate how sensitive $\epsilon$ is to repeated sea surface temperature (SST) and sea ice boundary conditions (AM4 Control), prescribed observed boundary conditions (AM4 PSST), and prescribed observed boundary conditions and prescribed forcing agents (AM4 PSST+ERF).

In CMIP6 Control (1293 realizations and 47 models), we compute $\epsilon$ as the $\pm 2\sigma$ range of all 1293 trends. We also compute $\epsilon$ for each of the 47 models and the multi-model mean results in the same estimate of $\epsilon$ (Supplementary Table 1). As in CMIP6 Control, in CMIP6 Historical (142 realizations and 5 models), the $\epsilon$ for all 142 trends is identical to the multi-model mean's estimate of $\epsilon$ (Supplementary Table 2).

In AM4 Control (100 realizations), the same SSTs and sea ice are prescribed every year with forcing agents at 1850 levels (see "Methods" section). This provides an estimate of $\epsilon$ in an atmosphere model in the absence of changing forcing agents, as well as changing SST and sea ice boundary conditions (AM PI).

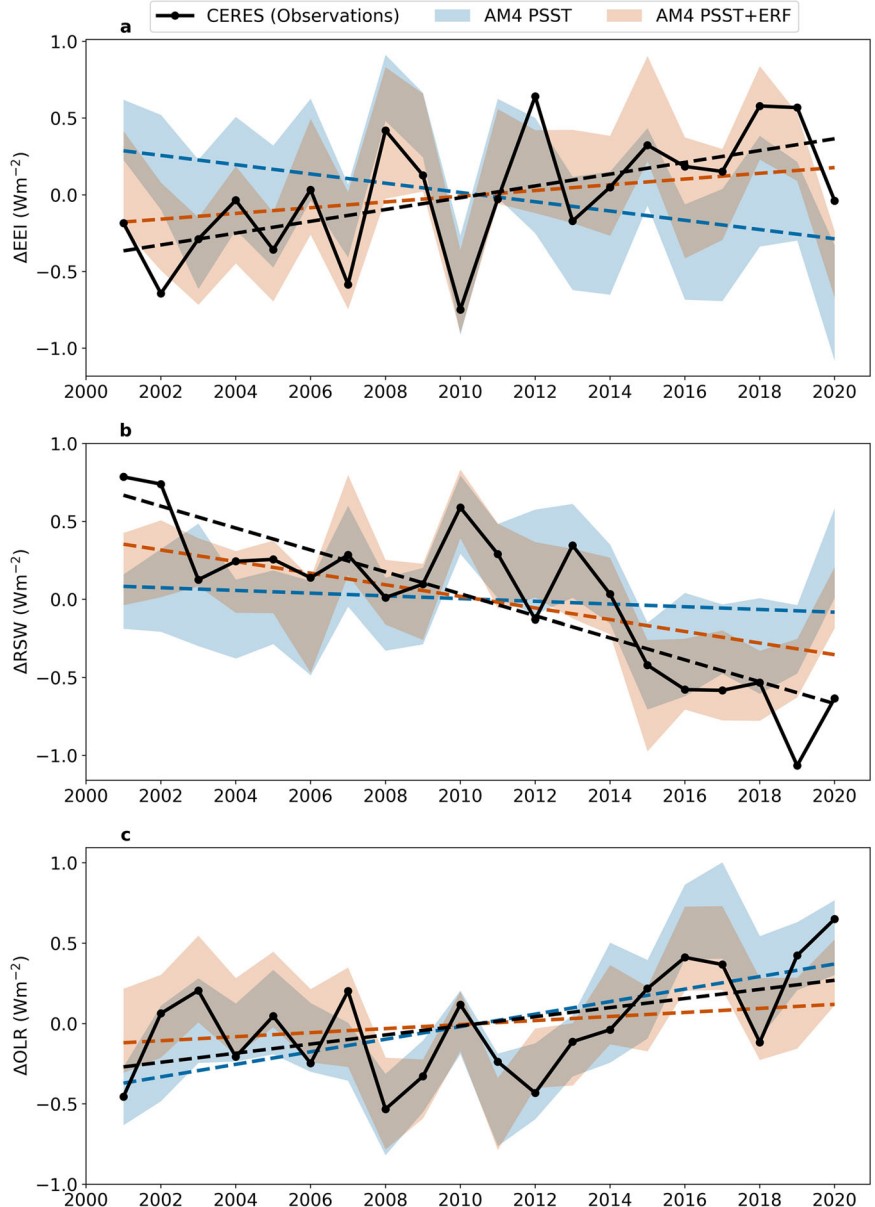

**Fig. 1 Global-mean Earth's radiation budget time series.** Interannual anomalies (denoted by Δ) in **a**, Earth's Energy Imbalance (EEI), **b**, reflected shortwave radiation (RSW), **c**, outgoing longwave radiation (OLR) during 2001–2020. Blue and orange shading each represent the full range of twenty-time series realizations in each ensemble. Positive (negative) values indicate more energy in the Earth system for EEI (RSW and OLR). Blue and orange dashed lines represent ensemble mean trends. CERES = Clouds and the Earth's Radiant Energy System satellite observations (black), AM4 PSST = Prescribed Sea Surface Temperatures (SSTs) and sea ice with forcing agents held fixed at 2014 levels in Geophysical Fluid Dynamics Laboratory Atmosphere Model 4 (AM4) (blue), and AM4 PSST+ERF = same as AM4 PSST but with effective radiative forcing changes (ERF; forcing agents varying) (orange).

In AM4 PSST (20 realizations), each realization has fixed radiative forcing, but prescribed SSTs and sea ice (AMIP PD estimate of $\epsilon$). Because forcing agents are held fixed at 2014 levels, the trend in force is 0, i.e., $\Delta$ERF = 0 in Eq. 1 and hence the ensemble mean provides the best estimate of the $\lambda\Delta T_S$ trend. It is important to note that this experiment contains imprints of a forced system (i.e., $\lambda\Delta T_S$) since SSTs and sea ice are prescribed but do not include the direct radiative impact of the forcing agent changes (i.e., $\Delta$ERF = 0) (see "Methods" section).

In the final experiment (AM4 PSST+ERF–20 realizations), in addition to prescribed SSTs and sea ice (observed), we also prescribe the CMIP6 forcing time series (well-mixed greenhouse gases concentrations, aerosol emissions, etc.) to obtain the sum of

the radiative forcing and response changes (see "Methods" section). Similar to AM4 PSST, the ensemble mean provides the best estimate of $\Delta$ERF + $\lambda\Delta T_S$ and the $\pm2\sigma$ range of the twenty trends provides the fifth estimate of $\epsilon$ (another AMIP PD estimate). It follows that the difference between AM4 PSST and AM4 PSST+ERF ensemble-mean trends provide an estimate of the trend due to the effective radiative forcing changes ($\Delta$ERF), i.e., the impact of changing forcing agents on the radiation budget (see "Methods" section).

The $\epsilon$ association with each of our 5 estimates is nearly identical at ~±0.19 Wm$^{-2}$decade$^{-1}$ over a 20-year period despite having very different boundary conditions and forcing conditions (CMIP6 Control ±0.19 Wm$^{-2}$decade$^{-1}$, CMIP6 Historical ±0.20

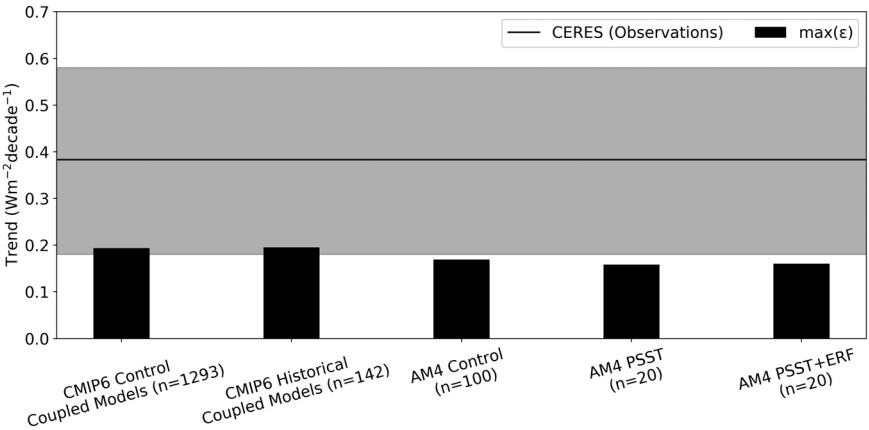

**Fig. 2 Global-mean observed trend in Earth's energy imbalance (EEI) is unexplained by internal variability.** Each model ensemble's estimate of maximum trends in ΔEEI due to internal variability (ε) is plotted. See Supplementary Tables 1–2 for the number of realizations (n; two-decade periods trends) for each model in Coupled Model Intercomparison Project Phase 6 (CMIP6) Control and CMIP6 Historical ensembles. Gray shading represents observational uncertainty. Note that although there is a small overlap between the high trends due to internal variability and the low trends due to observational uncertainty, the probability that both events happen in the same two-decade period is less than 1%; see text and see "Methods" section for details. CERES = Clouds and the Earth's Radiant Energy System satellite observations (black), AM4 PSST = Prescribed Sea Surface Temperatures (SSTs) and sea ice with forcing agents held fixed at 2014 levels in Geophysical Fluid Dynamics Laboratory Atmosphere Model 4 (AM4), and AM4 PSST+ERF = same as AM4 PSST but with effective radiative forcing changes (ERF; forcing agents varying).

$Wm^{-2}decade^{-1}$, AM4 Control $\pm 0.17\ Wm^{-2}decade^{-1}$, AM4 PSST $\pm 0.16\ Wm^{-2}decade^{-1}$, AM4 PSST+ERF $\pm 0.16\ Wm^{-2}decade^{-1}$). The similar ε value in trends is not seen in the amplitude of interannual anomalies, where variability in ΔEEI increases with the inclusion of SST variability (e.g., between AM4 Control and CM4 Control (1 of the 47 models in the CMIP6 Control ensemble)) (Supplementary Fig. 4). The larger amplitude interannual fluctuations of opposite signs cancel in the computation of a linear fit, yielding the same ε in EEI trends across experiments with different boundary conditions. The fact that ε is not altered between AM4 PSST and PSST+ERF ensembles, as well as between the CMIP6 Control and CMIP6 Historical ensembles, suggests that ε is independent of forcing and feedbacks.

We find that the CERES-observed TEEI lies outside the range of trends driven by internal variability alone using two different uncertainty metrics. In the first metric, we compare CERES (with its observational uncertainty), a single realization, to the ±2σ range of trends (ε). This ε is the model ensemble-generated uncertainty due to internal variability (Fig. 2 and Fig. 3a). The CERES TEEI ($0.38 \pm 0.20\ Wm^{-2}decade^{-1}$; uncertainty given by observational uncertainty) lies outside of our estimates of internal variability ($\epsilon \sim \pm 0.19\ Wm^{-2}decade^{-1}$) no matter which of our five experiment ensembles we estimate it from (Fig. 2). Although there is a small overlap between the lower end of the CERES observational range and max(ε) in Fig. 2, the probability that CERES has a low trend due to drift and that there was a high trend due to internal variability in the same two-decade period is extremely small (<1% probability) (see "Methods" section). Thus, the probability that internal variability could have caused the observed trend, even in the presence of large observational uncertainty, is exceptionally unlikely[42].

The value of ε is robust across 47 different CMIP6 Control models (Supplementary Table 1). One of the 47 models, GFDL CM4 Control ($\epsilon = \pm 0.20\ Wm^{-2}decade^{-1}$), is representative of the CMIP6 Control ensemble and is further analyzed in Fig. 3 alongside the GFDL AM4 results to maintain consistency in the hierarchy of GFDL modeling experiments. We also tested multi-millennial control simulations in two older generation GFDL models and found similar values of ε (ESM2M Control and CM3 Control).

The second uncertainty metric compares the 95% CI around the mean and is a common method that has been employed in previous studies. To be consistent with the 95% CI uncertainty estimate attached to CERES, we compute the 95% CI for each model ensemble's mean. The CERES TEEI with its 95% CI ($0.38 \pm 0.24\ Wm^{-2}decade^{-2}$) is (1) greater than 0 and (2) greater than the AM4 Control and CMIP6 Control ensemble means with their 95% CI uncertainty estimates ($0.01 \pm 0.02\ Wm^{-2}decade^{-1}$ and $0.00 \pm 0.01\ Wm^{-2}decade^{-1}$, respectively) (Fig. 4, Supplementary Fig. 1, Supplementary Table 1). This result again implies that internal variability is extremely unlikely to have caused the observed TEEI.

We note that comparing the CERES TEEI with its 95% CI (second metric) to the ±2σ range of trends (ε; first metric) would overestimate the uncertainty since CERES's standard error component of the 95% CI uncertainty is also a measure of the range of trends that could have been obtained by variability in the climate system. In fact, an alternative 95% CI for CERES could be to replace the standard error component with the modeled internal variability as a broader estimate of internal variability uncertainty (Fig. 4). Hereafter, when we compare CERES with the model ensembles, we will follow only one of the two metrics at any given time: CERES (one realization) with observational uncertainty is compared to ±2σ range of trends (ε) (Fig. 2, Fig. 3) and CERES with 95% CI is compared to the model ensemble mean with 95% CI (Fig. 4, Supplementary Fig. 1). These results imply that it is exceptionally unlikely that internal variability caused the observed TEEI and therefore implies that trends in radiative forcing and the associated climate response caused the observed TEEI.

**Decomposition of EEI into radiative forcing and response trends.** Since ε comprises a range of trends in EEI due to internal variability, the value of ε, $\pm 0.19\ Wm^{-2}decade^{-1}$, informs us of the estimated contribution internal variability can make to TEEI. Following Eq. 1, this would imply that the CERES TEEI contains a positive $\Delta ERF + \lambda \Delta T_s$ trend, but with a large range of 0.11–0.65 $Wm^{-2}decade^{-1}$ (95% CI range using ε instead of standard error in the internal variability component of uncertainty; Fig. 4). The lower bound of this range represents the minimum contribution

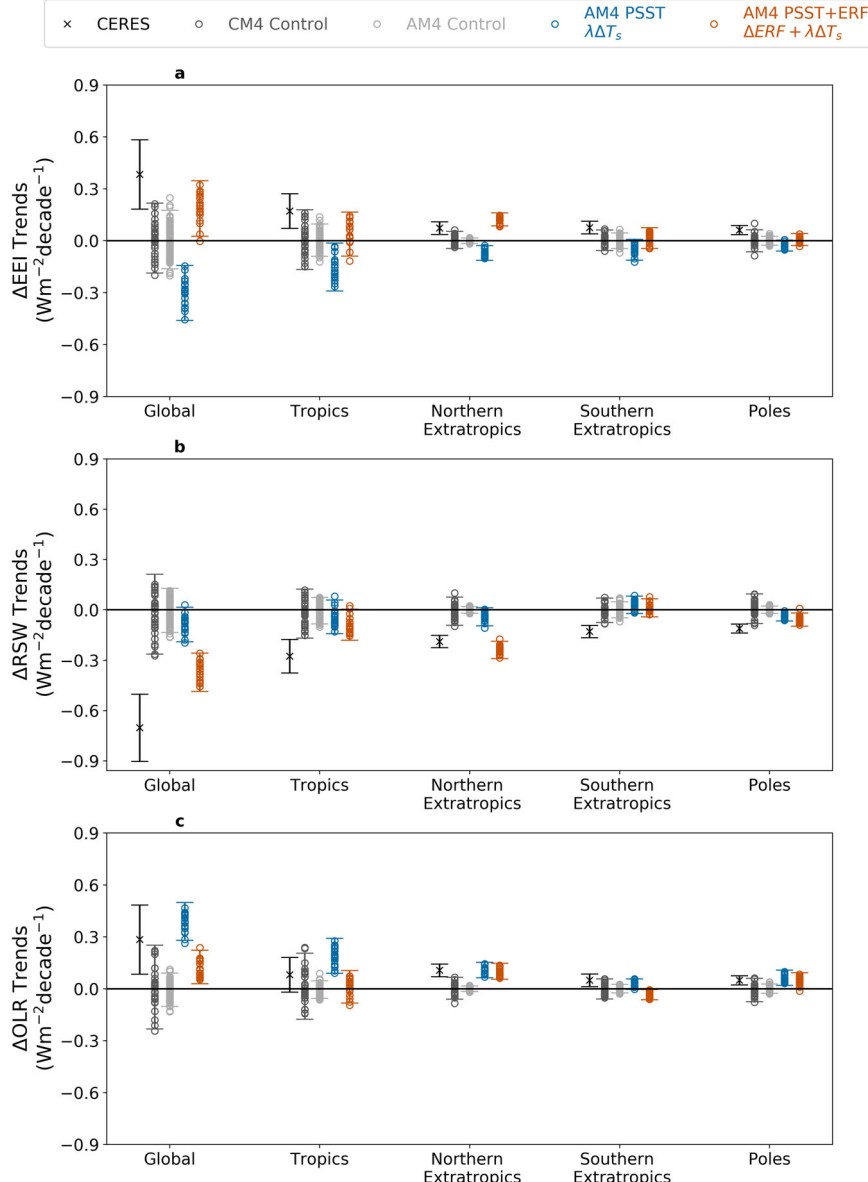

**Fig. 3 Regional radiative trends in Clouds and the Earth's Radiant Energy System (CERES) observations and a hierarchy of Geophysical Fluid Dynamics Laboratory (GFDL) climate model experiments. a** Earth's energy imbalance (EEI). **b**, Same as **a**, but for reflected shortwave radiation (RSW). **c** Same as **a**, but for outgoing longwave radiation (OLR). Positive EEI indicates more energy in the system while positive RSW and OLR indicate less energy. Values are relative to the global mean, i.e., area-weighted. Tropics (30°S–30°N), northern extra-tropics (30°N-60°N), southern extra-tropics (30°S–60°S), and poles (60°S–90°S and 60°N–90°N). $\pm 2\sigma$ range marked by dashes. For models, this is the range of radiative trends due to internal variability ($\epsilon$). For the observations, this is the range due to observational uncertainty. See Supplementary Fig. 1 for the equivalent figures but with 95% confidence intervals as uncertainty estimates and text for details. CERES's global trend in $\Delta$EEI exceeds control experiments' $\epsilon$ (see text for details and Fig. 2) and only lies in the realm of the experiment with prescribed SST and sea ice (observed) and effective radiative forcing (AM4 PSST+ERF). See Eq. 1 and text for details on $\lambda\Delta T_s$(blue) and $\lambda\Delta T_s+\Delta$ERF(orange) ($\Delta$ represents anomalies). CM4 = GFDL Coupled Model 4 (CM4 Control given by dark gray) and AM4 = GFDL Atmosphere Model 4 (AM4 Control given by light gray).

by anthropogenic forcing and response. Conversely, the upper bound of this range represents the maximum contribution by anthropogenic forcing and response. Indeed, we find that the ensemble mean estimate of AM4 PSST+ERF of $0.19 \pm 0.04$ Wm$^{-2}$decade$^{-1}$, is consistent with this range (Fig. 4, Supplementary Fig. 1a). All CMIP6 Historical coupled models analyzed in this study (only large ensembles were considered) lie in the CERES 95% CI range, implying that the anthropogenic forcing and response in coupled and atmosphere-only models yields the

observed positive trend (Fig. 4, Supplementary Table 2). We note that, while accounting for observational trend uncertainties, a lesser estimate of the satellite-related-trend-uncertainty than used here would further elevate the role that radiative forcing and feedbacks have played in the TEEI.

The AM4 experiments show that the positive trend can be attributed to a positive $\Delta$ERF trend (0.49 Wm$^{-2}$decade$^{-1}$; Table 1) overcoming a negative trend in $\lambda\Delta T_s$ ($-0.30$ Wm$^{-2}$decade$^{-1}$; Fig. 3a, Supplementary Fig. 1a). This forcing trend of AM4 is in

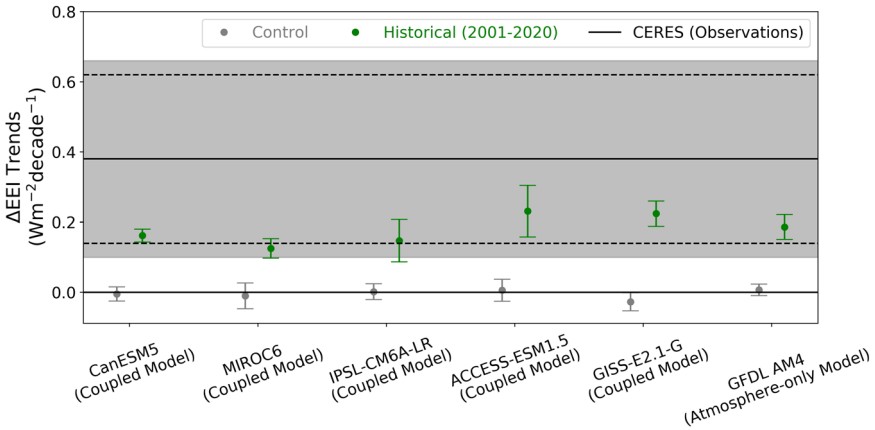

**Fig. 4 Global-mean observed trend in Earth's energy imbalance (EEI) obtained by anthropogenic forcing and the associated climate response.** Trends in ΔEEI (TEEI) with 95% confidence intervals (CI) (Δ represents anomalies). Dashed lines indicate Clouds and the Earth's Radiant Energy System (CERES) TEEI's 95% CI derived from observational uncertainty and standard error of linear fit (internal variability). Shading indicates CERES TEEI's 95% CI with internal variability uncertainty estimated from model-derived $\epsilon$. The lower end indicates the minimum contribution by anthropogenic forcing and the associated climate response ($\Delta ERF + \lambda \Delta T_s$; see Eq. 1) to CERES TEEI. Conversely, the upper end indicates the maximum anthropogenic contribution to CERES TEEI. Geophysical Fluid Dynamics Laboratory Atmosphere Model 4 (GFDL AM4) 'Historical' value represented by the experiment with prescribed sea surface temperatures and sea ice and effective radiative forcing changes (forcing agents varying) (AM4 PSST+ERF). Control experiments denoted by light gray filled circles. Historical experiments denoted by green filled circles. CanESM5 = Canadian Earth System Model version 5, MIROC6 = Model for Interdisciplinary Research on Climate version 6, IPSL-CM6A-LR = Institut Pierre-Simon Laplace-Climate Model 6-Low Resolution, ACCESS-ESM1.5 = Australian Community Climate and Earth-System Simulator Earth System Model Version 1.5, GISS-E2.1-G = Goddard Institute for Space Studies (GISS) climate model.

**Table 1 Decomposition of effective radiative forcing trends into greenhouse gas, aerosol, and natural radiative forcing contributions during 2001-2020.**

|  | All forcing | All forcing | GHG only | AER only | NAT only |
|---|---|---|---|---|---|
| Effective radiative forcing trend (Wm$^{-2}$ decade$^{-1}$) | AM4 ERF | AM4 RFMIP | AM4 RFMIP | AM4 RFMIP | AM4 RFMIP |
| ΔEEI | 0.49 ± 0.05 | 0.54 ± 0.10 | 0.40 ± 0.10 | 0.12 ± 0.10 | −0.03 ± 0.10 |
| ΔRSW | −0.28 ± 0.03 | −0.31 ± 0.07 | −0.12 ± 0.07 | −0.19 ± 0.07 | 0.07 ± 0.07 |
| ΔOLR | −0.26 ± 0.03 | −0.29 ± 0.05 | −0.27 ± 0.05 | 0.07 ± 0.05 | −0.10 ± 0.05 |

Uncertainty gave by 95% confidence intervals. Positive values of Earth's energy imbalance (EEI) trends and negative values of reflected shortwave radiation (RSW) and outgoing longwave radiation (OLR) trends indicate more energy into the system, respectively.
*RFMIP* Radiative Forcing Intercomparison Project, *GHG* greenhouse gas only, *AER* aerosols only, *NAT* natural forcing agents only.
Trends computed for anomalies time series (Δ).

excellent agreement with other CMIP6 Radiative Forcing Modeling Intercomparison Project's (RFMIP[43]) model trend estimates (MMM 0.50 ± 0.06 Wm$^{-2}$decade$^{-1}$; Supplementary Table 3). We have performed further experiments using the RFMIP GFDL AM4 ensemble to break down the trends in ΔERF due to anthropogenic well-mixed greenhouse gases (0.40 ± 0.10 Wm$^{-2}$decade$^{-1}$), anthropogenic aerosols (0.12 ± 0.10 Wm$^{-2}$decade$^{-1}$), and natural (−0.03 ± 0.10 Wm$^{-2}$decade$^{-1}$) (Table 1). Since the only notable positive trends come from the direct impact of anthropogenic radiative forcing, we conclude that the positive trends in CERES would not be possible without it. In the following section, we decompose TEEI into RSW and OLR trends and investigate their ΔERF and $\lambda \Delta T_S$ components.

**Decomposition of OLR and RSW into radiative forcing and response trends.** The CERES-observed trend in OLR is found to be 0.28 ± 0.22 Wm$^{-2}$decade$^{-1}$ (Fig. 3c, Supplementary Fig. 1c; radiation leaving Earth is positive for ΔOLR and ΔRSW). Our modeling experiments show that this is due to a near offset between a positive feedback component and a negative forcing component. When forcing is fixed, ΔOLR emitted over this period is 0.39 ± 0.02 Wm$^{-2}$decade$^{-1}$ (AM4 PSST; Fig. 3c, Supplementary Fig. 1c). AM4 PSST's prescribed boundary conditions use observed

SSTs that are increasing over this time period (observed global mean surface temperature trend of 0.23 ± 0.02 K/decade (NASA GISTEMP[44])) while well-mixed greenhouse gas concentrations are held fixed at 2014 levels, resulting in more infrared radiation emitted by Earth. This damping of the climate system in the absence of forcing is consistent with the dominance of the Planck and lapse rate feedbacks that alter OLR, which is Earth's primary way to lose energy to space.

When well-mixed greenhouse gas concentrations and other forcing agents are allowed to vary, the OLR trend decreases to 0.13 ± 0.02 Wm$^{-2}$decade$^{-1}$ (AM4 PSST+ERF, Fig. 3c, Supplementary Fig. 1c). This amounts to a trend in forcing of −0.26 Wm$^{-2}$decade$^{-1}$, entirely coming from well-mixed greenhouse gas changes (Table 1), which is consistent with previous work that studied trends in the greenhouse effect[45]. The reduction in OLR due to globe-wide increasing concentrations of well-mixed greenhouse gases is seen at most latitudes (Fig. 3c, Supplementary Figs. 1c, 5, 6). This is because $CO_2$ and other greenhouse gases' radiative trapping occurs in the infrared spectrum and acts to counter the loss of energy via the aforementioned feedbacks.

The CERES-observed trend in RSW is −0.70 ± 0.23 Wm$^{-2}$decade$^{-1}$ (Fig. 3b, Supplementary Fig. 1b). In comparison, the mean RSW trend of the AM4 PSST+ERF ensemble is −0.37 ± 0.03 Wm$^{-2}$decade$^{-1}$. The ΔERF component,

$-0.28\,\mathrm{Wm^{-2}decade^{-1}}$, dominates this reduction in reflection. The $\lambda\Delta T_S$ component, $-0.09\,\mathrm{Wm^{-2}decade^{-1}}$ acts to supplement this decrease in reflection and is discussed in detail later in this section.

Using the RFMIP GFDL AM4 ensemble, we find a near-equal contribution by aerosol forcing ($-0.19\,\mathrm{Wm^{-2}decade^{-1}}$) and greenhouse gas forcing ($-0.12\,\mathrm{Wm^{-2}decade^{-1}}$) in reducing reflection of shortwave radiation (Table 1). The aerosol forcing contributed to the RSW forcing component acts primarily in the northern extra-tropics (Fig. 3b and Supplementary Fig. 7e). We find a reduction in RSW in the northern extra-tropics consistent with an aerosol drawdown during this period over the northern extra-tropics (USA and Europe) (Fig. 3b, Supplementary Figs. 7–9)[46,47]. As aerosols decrease, there is less reflection of sunlight from the aerosols (direct effect) and from the clouds which have smaller optical depths and liquid water paths (indirect effect)[48]. Indeed, we find that satellite observations from MODIS[49] show a decrease in liquid water path over the northern extra-tropics, consistent with AM4 PSST+ERF liquid water path trends for that region (Supplementary Fig. 7c). The northern midlatitude aerosol forcing outweighs land-use forcing changes (Supplementary Fig. 7d, f). However, there is uncertainty regarding the CMIP6 aerosol emissions, which are used in AM4 PSST+ERF[47].

The greenhouse gas forcing contribution to the $\Delta$ERF component of the RSW trend manifests as rapid cloud adjustments. Increasing greenhouse gases cause a decrease in longwave cooling and an increase in atmospheric absorption which enhances tropospheric heating[50–54]. This reduces the relative humidity and hence reduces cloud cover[50,51]. These rapid cloud adjustments are independent of the $\lambda\Delta T_S$ component over the 20-year period discussed below, which is tied to surface temperature change. Together, approximately half of the $-0.28\,\mathrm{Wm^{-2}decade^{-1}}$ AM4-estimated RSW forcing trend is due to cloud changes (Supplementary Table 4).

Half of the AM4 PSST RSW trend ($\lambda\Delta T_S$ component) comes from the tropics (Fig. 3b, Supplementary Fig. 1b, Supplementary Table 5). This decrease in tropical reflection of solar radiation arises solely from trends in the prescribed SSTs. However, we do not find that the so-called pattern effect[55–65], the phenomenon wherein East-West SST gradients in the tropical Pacific modulates radiation and feedbacks, can explain this RSW $\lambda\Delta T_s$ trend (see "Methods" section). Two different metrics of Pacific SST gradients[59,66] show equal East Pacific and West Pacific warming trends, i.e., a neutral pattern, yet there is a negative modeled tropical RSW trend. Furthermore, CERES observations show a much stronger tropical RSW trend (Supplementary Figs. 10–11, Supplementary Table 6).

The other half of the AM4 PSST RSW trend comes from the northern extra-tropics. This is dominated by the clear-sky component (Supplementary Table 5) and is most likely due to land-use changes (Supplementary Fig. 7f). Over the poles and southern extra-tropics, we find considerable disagreement between CERES observations and the model (Fig. 3b). In the polar regions, Arctic and Antarctic sea ice is decreasing in CERES observations, which decreases reflection (Fig. 3b, Supplementary Figs. 7b, 12a). However, we find that the model, driven with AMIP boundary conditions, shows Antarctic sea ice to be increasing over this time period, leading to a smaller decrease in polar RSW in AM4 PSST. The AMIP sea ice and observation sea ice discrepancy is consistent with previous studies[35,67]. In the southern extra-tropics, the Southern Ocean cloud fraction is decreasing in CERES-MODIS observations while increasing in the model (Supplementary Fig. 12b, 13), which leads to more reflection in the model. After accounting for the polar, southern extra-tropical, and tropical discrepancies ($-0.33\,\mathrm{Wm^{-2}decade^{-1}}$), the model could match the observed global reduction in RSW. The model-observation discrepancies and uncertainty are further examined in the Discussion section.

## Discussion

The satellite-observed positive EEI trend over the 2001–2020 period is exceptionally unlikely (<1% probability)[42] to be explained by internal variability, which we estimate across 47 CMIP6 coupled models in preindustrial conditions, 5 CMIP6 coupled models in present-day conditions, and a hierarchy of GFDL atmosphere-only model experiments in preindustrial and present-day conditions. These results imply that the observed EEI trend meets the criteria for detection above internal variability and the simulated AM4 PSST+ERF trend also meets the criteria for emergence above internal variability (Figs. 2–3, Supplementary Fig. 14)[31,68]. Only by accounting for the temporal changes in anthropogenic forcing agents and the associated climate response in CMIP6 Historical coupled models and GFDL AM4 PSST+ERF was it possible to achieve trends that are in agreement with observations (Fig. 4). Thus, we conclude that the observed EEI trend is attributable to anthropogenic forcing and response[69].

The observed positive EEI trend shows that the heat uptake by the Earth system has accelerated over the past two decades. This occurred because Earth has gained energy at a faster rate than it has lost ($\Delta$ERF $> \lambda\Delta T_s$). Because of the role internal variability can play at short timescales, this rate of gain need not have significantly outpaced the rate of loss, i.e., display a detectable positive EEI trend. For example, Supplementary Fig. 15 shows that only by 2018 does the CERES-observed EEI trend emerge above internal variability, despite a positive ERF trend leading up to 2018 (Supplementary Fig. 16). This shows the value of satellite observations maintaining a climate data record, and further extending it will help reduce the uncertainty due to internal variability.

Radiatively, the significant positive trend in observed EEI is driven by a $-0.70\pm0.23\,\mathrm{Wm^{-2}decade^{-1}}$ trend in RSW and a $0.28\pm0.22\,\mathrm{Wm^{-2}decade^{-1}}$ trend in OLR. While the dominance of the RSW component of the EEI trend has been shown in future projections of climate change, our work shows, with direct observations, that this is already happening in the current climate[19,70].

Our study provides estimates of model uncertainty due to internal variability ($\epsilon$). When observations and the model do not agree even after accounting for internal variability, model biases could be the cause of the discrepancy. For example, our modeled RSW and OLR trends are mostly consistent with the observed trends, however, the model also shows some inconsistencies with the observations. First, the model reflects more sunlight than observations in the tropics during this period. This could arise from a lack of aerosol decrease in the tropics, weak greenhouse gas rapid cloud adjustment, or weak response to the underlying SST pattern. Second, in the extra-tropics and poles, the model displays more reflection of sunlight than observations, arising from discrepancies between observations and the model's land-use changes, excessive aerosol drawdown over China, CMIP6 prescribed sea-ice boundary conditions, and modeled Southern Ocean cloud cover. Third, the model traps more infrared radiation than observations due to the weaker modeled tropical longwave cloud radiative effect trend (Supplementary Figs. 1, 5–8, 12–13, Supplementary Table 7).

These results imply that, first, we need to understand why the 2001–2020 SST warming pattern caused tropical RSW to decrease (AM4 PSST). Existing tropical pattern effect theories only help explain monthly-annual variations in RSW but cannot explain the decadal trend (Supplementary Figs. 10–11, Supplementary Table 6; see "Methods" section). Second, aerosol-radiation interactions over the tropics (which may have contributed to the observed tropical RSW decrease) and China need to be better understood. Despite evidence for decreasing aerosol emissions and aerosol optical depths in observations in China in the latter half of this period[46,47], RSW increased in CERES instead of an

expected decrease (Supplementary Fig. 8a). Furthermore, understanding why GFDL AM4 and potentially other CMIP6 models[47] show a decrease in RSW, despite CMIP6-prescribed increasing aerosol emissions in China, is an avenue for further exploration. Third, we need to be cautious in using CMIP6 sea-ice data. Fourth, we need a better comprehension of the impact of clouds on Earth's radiation budget on decadal time scales. Finally, we find that CMIP6 Historical coupled models with historical forcing show similar trends to AM4 PSST+ERF, but both lie at the lower end of the CERES range (Fig. 4). Future work could aim to explain whether this arises because of weaker forcing or a strong radiative response or both in models during this 2001–2020 period.

## Methods

$\Delta$EEI represents monthly anomalies and is calculated by subtracting the long-term monthly mean EEI of the period, from each year's month. For example, $\Delta EEI_{Jan,2020} = EEI_{Jan,2020} - \overline{EEI}_{Jan,2001-2020}$. Trends were then calculated by obtaining the slope of the linear fit through the anomalies time series. Regional values were calculated relative to the global mean, i.e., area-weighted: Poles (90°S −60°S and 60°N−90°N; 13.4% of Earth's area), Southern Extra-Tropics (60°S −30°S; 18.3% of Earth's area), Northern Extra-Tropics (30°N−60°N; 18.3% of Earth's area), and Tropics (30°S−30°N; 50% of Earth's area). $\epsilon$ is the $\pm 2\sigma$ range of trends in each model ensemble experiment.

**Models.** In the CMIP6 Control experiments, boundary conditions were allowed to vary in multi-century piControl simulations using coupled models with forcing fixed at 1850 levels. We analyzed 47 CMIP6 coupled models' piControl simulations to estimate $\epsilon$. We use 1 realization (r1i1p1f1) per model and slice the piControl time series into consecutive, non-overlapping 20-year periods. The $\pm 2\sigma$ spread of all these trends (1293 realizations) is our estimate of $\epsilon$ for CMIP6 Control. Furthermore, for each model, we calculate $\epsilon$. The multi-model mean $\epsilon$ is identical to the $\epsilon$ obtained by computing the $\pm 2\sigma$ range for all 1293 trends (Supplementary Table 1).

Apart from GFDL CM4 Control, none of the models had significant drift, so we use the full-time series available. In the case of CM4 Control, 1 of the 47 models, the simulation had 650 years of data, but we exclude the first 20 years due to drift (Supplementary Fig. 17a–c). The trend through the remaining years' time series is negligible, $-0.002 \pm 0.007$ Wm$^{-2}$century$^{-1}$, implying that it can be used for the analysis. The mean EEI (TOA imbalance) is 0.28 Wm$^{-2}$. See Supplementary Table 1 for model names, number of periods (i.e., realizations), trends, and $\epsilon$ values. In addition, we used multimillennial simulations[71] from two older generations of GFDL models (ESM2M and CM3) to study the trends in EEI over hundreds of consecutive 20-year periods.

In CMIP6 Historical, we analyzed the 5 available CMIP6 coupled models that contained at least 10 realizations per model. This provided us with single model large ensembles with historical forcing (till 2014) and SSP2-4.5 forcing (2015–2020) in coupled models during the CERES era period January 2001–December 2020. See Supplementary Table 2 for model names, number of realizations, trends, and $\epsilon$ values. CanESM5 had a 25-member ensemble with 'p1' physics and another 25-member ensemble with 'p2' physics[72]. Both of these ensembles individually fall into the CERES 95% CI range in Fig. 4. GISS-E2.1-G had a 10-member ensemble with 'p1f2' configuration, a 5-member ensemble with 'p3f1' configuration, and a 4-member ensemble with 'p5f1' configuration[73,74]. Each of these ensembles falls into the CERES range in Fig. 4 (see also Supplementary Table 2).

The AM4 Control experiment was conducted by providing GFDL AM4 with the SST pattern of a GFDL CM4 piControl simulation (CM4 Control). This SST pattern was then repeated year after year for 200 years as the boundary condition for GFDL AM4. We then randomly picked years from the 200-year EEI time series and made a 2000-year time series (bootstrap). Next, 100 20-year periods (Supplementary Fig. 17d) are sampled consecutively. These 100 20-year periods provide 100 linear trends. The $\pm 2\sigma$ spread of these 100 trends is our estimate of $\epsilon$ for AM4 Control. Forcing was fixed at 1850 levels.

In AM4 Fixed Forcing, we prescribed observed SSTs and sea ice over 2001–2020 in GFDL AM4 (AMIP-style[75]). These AMIP simulations used the monthly SSTs and sea ice concentrations prepared for the CMIP6 historical AMIP simulations[75,76], which were extended to December 2020 using the NOAA Optimum Interpolation (OI) SST V2 data[77]. Further details regarding the prescription and forcing datasets are listed in the GFDL AM4 model description (Appendix A)[11]. We created an initial condition large ensemble[31,68,78] with 20 realizations of Earth over this time period with forcing agents fixed at 2014 levels. The $\pm 2\sigma$ spread of these 20 trends provides our estimate of $\epsilon$ for AM4 PSST (units: Wm$^{-2}$ decade$^{-1}$). The year 2014 is used since it marks the end of CMIP6 emission prescriptions. Since we are studying the trends in an interannual time series, the particular year at which we fix forcing agents does not matter. It is important to note that although forcing was fixed, the boundary conditions were already forced

since they were prescribed. The warming trend has an impact on the top-of-atmosphere radiation budget, e.g., OLR increase.

Finally, in AM4 PSST+ERF, we repeated the AM4 PSST experiment but allowed forcing agents to vary over this time period, producing another 20 realizations. The $\pm 2\sigma$ spread of these 20 trends provides our estimate of $\epsilon$ for AM4 PSST+ERF. Forcing agents' variations follow CMIP6 emissions until 2014 and scenario SSP2-4.5 thereafter[79]. Each realization in AM4 PSST and AM4 PSST+ERF had the same boundary conditions but slightly different initial conditions, which yielded the so-called butterfly effect[80,81] that caused the climate state in each realization to be different from one another.

The decomposition of the effective radiative forcing trends over this period were calculated using the GFDL AM4 3-member ensemble in the Radiative Forcing Modeling Intercomparison Project (RFMIP). There were four temporal evolutions with different forcing agents: greenhouse gas only, aerosol only, natural forcings, and all forcings. These experiments were prescribed with AM4 Control boundary conditions (SST and sea ice) and forcing agents were allowed to vary. Uncertainties given in Table 1 were calculated by using the standard deviation ($\sigma$) of the trends in the AM4 Control experiment instead of using the small 3-member ensemble of the RFMIP experiments in order to get a more representative uncertainty. Differences in forcing estimated from (AM4 PSST+ERF−AM4 PSST) and RFMIP are negligible because they both use the same CMIP6 forcing-agent time series and only differ in boundary conditions (fixed SSTs for RFMIP and varying SSTs for AM4 PSST+ERF) (Table 1).

**Observations.** We used observations of top-of-atmosphere radiation from the Clouds and the Earth's Radiant Energy System Energy Balance and Filled (CERES EBAF Ed4.1). MODIS data from January 2001–July 2002 was obtained using the Terra platform alone and data from July 2002-December 2020 was obtained using an average of Terra and Aqua platforms. MODIS data poleward of 55° was ignored due to insufficient coverage (<19% of Earth's surface area). MODIS LWP data is pixel-weighted.

We omit 2000 in the analysis because CERES does not provide 2000's January and February data. To compare complete years (Jan-Dec), we begin with 2001. The significant positive trend in EEI in CERES EBAF 4.1 was not seen in an earlier data product version: CERES EBAF Ed2.8. This is because there was a smaller reduction in RSW in Ed2.8. This was mostly due to diurnal corrections used in Ed2.8 and not because of calibration differences[17,37].

CERES observational uncertainty in the absolute magnitude of fluxes arises from instrument calibration uncertainty, EBAF diurnal correction, and radiance-to-flux conversion[17]. However, since our study focuses on decadal trends, observational uncertainty due to the stability of the CERES instruments is more important than absolute accuracy.

Although CERES is highly stable, systematic errors in observational uncertainties in decadal trends could still arise and so we assume a 0.20 Wm$^{-2}$ decade$^{-1}$ observational uncertainty in EEI, RSW, and OLR at every grid point. The total probability that internal variability could have caused the global-mean observed TEEI, even in the presence of large observational uncertainty, is less than 1% because the right-end tail of the internal variability probability distribution is being multiplied by the left-end tail of an assumed normal distribution of the observational uncertainty. Moreover, this result is insensitive to the probability density function (normal distribution, left-skew normal distribution, or uniform distribution) that is assumed for the observational uncertainty. The two probabilities can be multiplied to obtain the total probability because they are independent events. Further uncertainty details in CERES and MODIS are outlined in their data product manuscripts[17,49].

**Confidence intervals.** 95% confidence intervals are listed for observations, as well as for model ensembles. For observations (only 1 realization), this 95% CI $= 1.96 \times \sqrt{\sigma^2_{Obs.} + \sigma^2_{Var.}}$, where $2\sigma_{Obs.} = 0.2$ Wm$^{-2}$ decade$^{-1}$ and $\sigma_{Var.}$ is the standard error associated with the linear fit[82]. The standard-error derived uncertainty[17,30,34,35] in CERES, an uncertainty that represents internal variability uncertainty, is similar in magnitude to the model-generated internal variability uncertainty ($\epsilon$).

Therefore, we also apply the model-derived $\epsilon$ as another way of estimating the 95% CI on CERES $\left( TEEI \pm 1.96 \times \frac{\sqrt{\sigma^2_{Obs.} + \left(\frac{\epsilon}{2}\right)^2}}{\sqrt{1}} \right)$. These uncertainties ($\sigma_{Obs.}$ and $\sigma_{Var.}$) are added in quadrature because they are independent uncertainties. For model ensembles, which contain multiple realizations, we make use of the spread in trends (linear fits) in the ensemble. The model ensemble mean confidence interval is calculated as: $1.96 \times \frac{\sigma}{\sqrt{n}} \sim \frac{\epsilon}{\sqrt{n}}$, where $\sigma$ is one standard deviation of trends in the ensemble and $n$ is the number of realizations in the ensemble. Therefore, the model ensembles in Supplementary Fig. 1 are identical to Fig. 3 but scaled by $\frac{1}{\sqrt{n}}$.

**Pattern effect.** Past work has suggested that East–West SST gradients in the tropical Pacific can modulate radiation and feedbacks, i.e., the pattern effect[55–65]. When the West Pacific is anomalously warmer than the East Pacific, the moist adiabat changes for the rest of the tropics too (weak temperature gradient[83]). This in turn would lead to a more negative lapse rate feedback and a more stable atmosphere in subsidence regions, yielding more low clouds and a more negative cloud feedback. Conversely,

when the East Pacific is anomalously warmer than the West Pacific, there is a more positive lapse rate feedback and a more positive cloud feedback. This would lead to a more positive total climate feedback and hence a larger EEI.

We hypothesize that this pattern effect would be observable over a 20-year period with the unforced simulations (CM4 Control), the forced simulations (AM4 PSST), and the satellite observations (CERES). Year-to-year forcing changes were removed in the latter dataset using the estimate of forcing from our experiments (Supplementary Fig. 10a; the difference between AM4 PSST+ERF and AM4 PSST). West Pacific and East Pacific boxes are drawn in Supplementary Fig. 13a.

We find that on an interannual basis, indeed, global $\Delta$RSW decreases as the East Pacific warms more than the West Pacific (Supplementary Fig. 11a, Supplementary Table 6). This relation is stronger for the tropics (Supplementary Fig. 10a; Supplementary Table 6). The results are insensitive to two different tropical Pacific SST gradient metrics[59,66]. These results are confirmed by the fully coupled model results (Supplementary Fig. 10a, gray circles). However, over the 2001–2020 period, the trend in the E–W $\Delta$SST gradient is flat, yet the trends in RSW are negative, contrary to the hypothesis (Supplementary Fig. 10b). CERES lies outside the lower end of this range, which indicates that the observations have a larger negative trend in RSW than the model for the same SST pattern trend. Finally, the pattern effect predicts a negative relationship between OLR and the E-W SST gradient. However, this is not what we find (Supplementary Fig. 11e). We find an increase in tropical all-sky and clear-sky $\Delta$OLR in most datasets (statistically significant for CM4 and AM4 but not for CERES–Supplementary Table 6).

## Data availability

CERES EBAF Edition 4.1 data was retrieved from the NASA Langley Research Center Atmospheric Science Data Center. MODIS data was obtained from The Level-1 and Atmosphere Archive and Distribution System (LAADS) Distributed Active Archive Center (DAAC). The observed surface temperature was obtained from NASA GISTEMP: GISTEMP Team, 2020: GISS Surface Temperature Analysis (GISTEMP), version 4. NASA Goddard Institute for Space Studies. Dataset accessed 2021 at https://data.giss.nasa.gov/gistemp/. The GFDL AM4 data generated in this study have been deposited in the Zenodo database under accession code https://doi.org/10.5281/zenodo.4784726[84]. CMIP6 Control and CMIP6 Historical data were obtained from the CMIP6 archive.

## Code availability

Code can be accessed at: https://doi.org/10.5281/zenodo.4784968[85].

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

## Acknowledgements

We acknowledge the World Climate Research Programme, which, through its Working Group on Coupled Modelling, coordinated and promoted CMIP6. We thank the climate modeling groups for producing and making available their model output, the Earth System Grid Federation (ESGF) for archiving the data and providing access, and the multiple funding agencies that support CMIP6 and ESGF. Numerical simulations were conducted with GFDL computational resources. We thank Pu Lin for helping extend the AMIP simulations. We thank Mike Winton, Yi Ming, Isaac Held, and Leo Donner for reviewing an earlier draft. S.P.R. was supported by the Future Investigators in NASA Earth and Space Science and Technology award 80NSSC19K1372 and partially supported by the High Meadows Environmental Institute at Princeton University through the Mary and Randall Hack '69 Research Fund.

## Author contributions

S.P.R. performed analysis and writing with regular feedback and inputs to the manuscript from D.P. and V.R. The AM4 simulations were performed by D.P.

## Competing interests

The authors declare no competing interests.
