## [Peer Review File · Nature Communications]

REVIEWER COMMENTS

Reviewer #1 (Remarks to the Author):

Raghuraman et al. provided a model-based analysis for the positive trend in EEI since 2000, they found that an increase in anthropogenic greenhouse gasses forcing is mainly responsible for this EEI increase, associated with changes in both ORL and RSW. Regional changes were marginally analyzed in this study. The EEI and its trend are of great importance in climate change, so the topic is relevant and intriguing. The paper is well-written and deserves publication in nature communications. I have several concerns that need to be addressed before publication.

Comments:

(1). Page-4, section 1 “Trends in CERES observations”. A significant trend in EEI was identified here. However, the CERES data was adjusted by ocean observations for its mean values, only the year-to-year variation is proved to be reliable (see many studies by Dr. Loeb and its group). This adjustment and its impact need to be discussed and assessed. Furthermore, there is a positive trend in recent CERES data, not in previous versions, so I wonder how reliable the trend is for CERES.

(2). Page-4, section 1 “Trends in CERES observations”. As ~90% of EEI was stored in the ocean (AR5), did ocean observations also show a rate increase?

(3). Page-6, line-115. Why 2014 was chosen?

(4). Figure 1b and associated discussions: why dRSW is negative in all regions? Why dOLR is positive in all latitude regions? More regional analyses are needed to better understand the results. Some geographical maps for model and observation will be appreciated.

(5). Line-191-192. It seems not uniform to me, at least not for Northern Extra-tropics. Again, some spatial maps were needed to better understand the changes.

(6). I have some difficulty in understanding “fixed forcing” experiment, why 2014 was chosen, when the EEI has already increased? Why not use the first year 2000 or 2001 as a baseline? Also the physical meaning of the difference between AM4 Varying Forcing and AM4 Fixed Forcing is also not very clear to me (i.e. Line 189).

(7). Maps for the spatial patterns of the All-forcing, GHG-only forcing, AER-only and NAT-only experiments will be very helpful to illustrate their contributions.

(8). Page-9, section “Estimates of forcing and feedback trends, 2001-2018”. There are many convoluted discussions, which needs to be better presented. I suggest to either shorten this section or divide it into two sections.

(9). Page-12, lines 221-229. This paragraph is to discuss the contradictions between models and observations. However, it is still not clear how the model uncertainty can impact the main

conclusion, which needs an explicit exploration. A comprehensive validation of model simulation (Varying Forcing experiment) against observations can be provided in the supplementary material.

Reviewer #2 (Remarks to the Author):

Review of Anthropogenic forcing yields significant positive trend in Earth's energy imbalance by Shiv Priyam Raghuraman, David Paynter , V. Ramaswamy

The manuscript by Raghuraman and colleagues seeks to evaluate the trends in TOA fluxes (EEI, RSW, OLR) using CERES EBAF 4.1 and identify the drivers of the trends using various simulations from GFDL AM4/CM4. The study concludes that anthropogenic forcing plays a key role in observed trends (though it contradicts itself regarding the details) and the regional analysis conducted in the manuscript draws links to the role of sea ice loss (which is known), aerosol trends (which are prescribed), and the tropical pattern effect on changes in albedo (which is also known). The manuscript is fairly well written and the experimental design description is clear. I have major concerns with the interpretation of the simulations and question whether the reasoning and statistical analyses are adequate, and these concerns are detailed below. I also question what is new here, as evident from the comments in parentheses above and suggest that any revised manuscript be explicit regarding findings that are both new and significant. Included are also various minor concerns regarding the lack of important information in the captions/text and the inclusion of excessive detail in others.

Major Comments

- There seem to be several internal contradictions in the abstract. For example, the abstract states that the observed trend is only achieved through anthropogenic greenhouse gases. This is odd as Table 1 shows this explicitly not to be the case. Rather aerosols are also fundamental. Moreover, the abstract then goes on to cite aerosol drawdowns as a key part of the TEEI. It then also identifies important contributions from internal variability associated with patterns of tropical warming. Also of this seems inconsistent.
- The interpretation of Figure 1 seems simplistic as it is framed as trends “with forcing” and “without forcing”. In fact there are vestigial characteristics of forcing in the “without forcing” as this simulation uses observed SST and sea ice. The authors make no effort so far as I can tell to account for these effects and it is unclear if they are fully aware of them. To do this without such contamination one would need fully coupled runs run with/without transient forcing - which would have the effect of increasing the noise and reducing the S/N. Some of the results might not be statistically significant once that is done but I would suggest that is a real hurdle and is what has potentially kept other authors from publishing similar analyses.
- Observational uncertainties and particularly the calibration stability uncertainty in CERES has not been adequately accounted for. The challenge is that the design calibration stability of

CERES is only slightly less than many of trends being identified. In truth, this uncertainty poses a major challenge for what the authors are trying to do yet it barely gets mention. This is problematic.

- Lastly, most of the conclusions in the manuscript are derived from a single model - yet the authors make almost not case for whether the model is suitable. In this context, there seem to be some clearly disagreement regarding regional model trends and those observed. This should concern the authors. Yet model evaluation doesn't seem to be addressed at all. In my opinion, it needs to be. At minimum, the model's ability to represent CERES mean states and seasonal cycle should be shown to be among the better models given how poorly we know some models to fair in reproducing observed trends. There are also important questions about forcing. The authors state that CMIP6 forcings are used - but don't say which ones. Various groups use different forcings based on the needs of their model (e.g. volcanic). Also, the CMIP6 historical era ends in 2014. What is used after 2014, particularly for aerosols (including biomass burning aerosols)? This is a first-order question in my view but I don't see any discussion.

Line By Line Comments

29: Why do you conclude the driver is anthropogenic greenhouse gases? This is not what the results show.

42: Sentence begins with a number. Please revise

64: It is not possible to separate contributions directly from observations but this statement suggests it is. Please revise.

74: This is not quite right. The simulations you are conducting are AMIP simulations. There are various aspects that are prescribed and thus the question is raised as to what aspects of these are forced and what are internal. There needs to be further discussion around this and I think there are implications for the broader interpretations in the manuscript. For example, if sea ice is prescribed and it is a component of the RSW trend, why do your no-forcing runs not show the associated RSW trend? The same can be argued of any SST forced changes and their associated cloud responses. I thus don't find the experiments to serve as a suitable contrast between "forced" and "unforced" states, as the authors would like to make them out to be. The discussion of the methods (line 317) seems to recognize this issue with regard to GHG but not other aspects. A complete, comprehensive discussion that addresses these issues and clearly articulates the points being made regarding figure

1 is needed. I am not convinced that a sound argument, making the points the authors seem to want to make, is justified.

Fig 1: The authors omit CERES data from 2000, with no justification and with the effect of increasing the magnitude of the estimated trend. They should not. They also should extend the record to be as up to date as possible for the purposes of the discussion.

Fig 1: There is no indication on the figure or in the caption what the shading represents. Std err? Std dev? Full range? This should be in the caption. Shouldn't the standard error be shown to make statements regarding the role of forcing and the 2-sigma range used to show the consistency with observations?

Fig 1: Does not say how regions are defined. Needs to - or state "see text". But seems to make more sense to put in the figure and remove from text.

Fig 1: CERES EBAF4.1 is available into Apr 2020. The data needs to be updated, even if to provide context for the periods of the runs in the discussion.

Fig 1: In order to make statements that are significant, a two std error range needs to be shown and used for the discussion. 1 std error signals are not significant. And why are they being referred to as error bars? And how does one compute standard error from a single member (i.e. obs)? Are these just in regarding to the trend fit? Clearly this is a significant underestimate of the uncertainty in trends.

Fig 1: I find Fig 1b to be redundant with Fig. 2. There is no need for Fig 1b so far as I can tell.

Fig 1: I'm confused by the authors interpretation of Figure 1 and I feel it is not as straight forward as the authors present. Part of the argument is that the climate response in sea ice and SST are what drive EEI. But aren't those included in AM4 Fixed Forcing? Arguably the lack of greenhouse gas increases the Planck response, the imposed sea ice should reproduce the RSW influence, the SST should drive changes in clouds and water vapor (and thus OLR). And so I don't think it is fair to interpret the 'fixed forcing' as being devoid of the impacts of forcing.

92: So the key question is what drives the RSW reduction contrast between the runs. It is not some of the key feedbacks, like sea ice, since both are prescribed. If it is forcing, this is pretty unsurprising as the reductions are prescribed. So the key question to me, and the one that needs to be clearly answered to make this a high profile publication, is whether the difference is driven by cloud responses to warming (outside of the Arctic). I don't think that question has been addressed.

126: I think you'll find a wide inter-model spread in TEEI variability when you go beyond GFDL models. Need to investigate further such as in the CMIP6 archive.

134: But it is almost certainly dependent on model.

135: Do you find this in the model, the observations, or both? Is it true of models generally?

Figure 2: It is stated that the error bars on CERES are the standard error? I don't understand how you compute a standard error with 1 member (obs) since standard error should be the standard deviation divided by the # of members minus one? Perhaps this relates to errors in the trend fit. If it does, I would clarify this further. For the model experiments, why is the standard deviation the relevant metric? Shouldn't the standard error also be shown to make the points that the forced signals are different between the runs (with the std dev used to make the point that CERES observations lie within the model spread).

146: Epsilon is not a single value but a $f(t)$ and so it can't be compared to TEEI. I suppose you mean the it is double the largest estimate of the contribution of epsilon to TEEI? The wording needs to be made more precise .

150: Assuming your estimate of internal variability is correct.

150: I don't think this is correct. The 0.4 value is the two-sigma range, not the 2-standard error range. Uncertainty in the forced response goes as the 2-standard error range.

161: What range is this? Std error or deviation?

164: Again, what range is this? Standard deviation is not the relevant measure.

168: This isn't a prediction.

173 and Fig. 2: The trends of the varying forcing for the NH extra tropics don't seem to be consistent with the observations for RSW or EEI since the observed value doesn't fall in the ensemble range. Shouldn't this be a cause for concern?

There is a lot of language in the figure captions dedicated to explaining the sign of the flux. State it once and be done with it. Rather include more useful information such as what exactly is on the plot.

197: Given this I am perplexed by the claim in the abstract that “this trend is achieved only upon accounting for the increase in radiative forcing by anthropogenic greenhouse gases,”

249: Is the gradient in warming the intended index (as stated) or the gradient in SST (as stated in line 248)?

Fig 3: What is the timescale here? Are these annual mean anomalies? How is the ensemble mean therefore plotted? Do you mean the ensemble mean annual means? This is the kind of information that I’m referring to above in saying more useful information needs to be in the captions.

Fig 3: Why isn’t the mean of the East-West SST difference negative? Surely some removal of the mean has been conducted - yet nothing is mentioned.

252: What is the uncertainty in the model estimated pattern effect? Surely this is fairly model dependent?

Fig 3: Is the point here that the annual means of the CM4 control run exhibit a relationship between the zonal gradient and tropical albedo? And that the control run does as well? Shouldn’t the regressions be computed and plotted? and their significance be more rigorously addressed? Is the point in b) that a similar relationship is expressed in trends in albedo and SST gradient?

Again a significant amount of text in the caption is dedicated to describing simple sign conventions and not converting important material about the figure such as these aspects.

277: “estimate” present tense.

308: There is no discussion of drift in the coupled control. Is it zero? What is the TOA imbalance?

311: Are the ESM2M or CM3 used anywhere in the manuscript? I see there is mention of trends in the PI Control - but really a more comprehensive estimate from CMIP6 should be used if the goal here is to show model robustness. These data are freely available.

314: Would be useful to state what datasets are used for the CMIP6 AMIP runs rather than reference other manuscripts.

319: The upshot is that your fixed forcing runs have several contrasting trends that are difficult to interpret. Some are associated with the forced response in imposed observations (SST, sea ice and associated changes) and some associated with internal variability. This makes the runs difficult to interpret.

343: A statement regarding the estimated calibration stability of CERES is warranted in addition to a statement regarding the consequences for the results and its basis.

347: Standard errors of the linear fit are only a subset of the uncertainties. Should the calibration stability uncertainties in CERES also be considered?

Table 1: This is a decomposition of the contribution to the trends, not the absolute forcing. This should be clarified.

The effective radiate forcing trend is only slightly larger than the change in EEI. This seems odd given that the present day imbalance is considerably less than the forcing (~50%). Why is this?

Reviewer #3 (Remarks to the Author):

“Anthropogenic forcing yields significant positive trend in Earth’s energy imbalance” by Raghuraman et al.

The content of the paper is interesting and useful for the EEI study, since the authors have tried to find the contributing drivers for the TEEI. The idea is quite novel and the method should be straightforward. However, I found it difficult to follow because of the vague description of the method details. I suggest to return to the authors for further revision.

Major comments

As I understand, in AM4 control, the years are randomly sampled to form 18-year time series, and in CM4 control run, the consecutive 18 year period is sampled by randomly selecting the start year.

Do you get ε by fitting Eq (1)?

It is not clear how ΔEEI is calculated. I can guess ΔT_s is calculated from monthly anomaly, but not ΔEEI .

Or the ΔEEI trend in the control run is regarded as the ε trend?

Minor comments

Lines 32-33: “The decrease in reflected solar radiation is due to midlatitude aerosol drawdown, spatial pattern changes in tropical warming, and polar sea-ice decrease” This is well known fact, not the new finding.

Line 48: It is not clear how the “interannual anomaly” is defined?

Line 50 : “can expressed” should be “can be expressed”

Is ΔEEI the EEI monthly anomaly?

Line 73: “This provides a 2σ range of trends as our estimate of”, what does this mean? How do you calculate it? Please give details in the method.

Figure 1a:

It would be better to show fitted lines for other two cases as well.

It is not clear how you get the error bar? Do you get the fitted slopes first, and then get the mean and STD of the slopes next?

Lines 120-121: “We estimate ε by computing the 2σ spread of trends in each ensemble (Figure 2).” It is not clear how ε is estimated.

Responses to Reviewers' Comments on "Anthropogenic forcing yields significant positive trend in Earth's energy imbalance"

Reviewer #1 (Remarks to the Author):

Raghuraman et al. provided a model-based analysis for the positive trend in EEI since 2000, they found that an increase in anthropogenic greenhouse gasses forcing is mainly responsible for this EEI increase, associated with changes in both ORL and RSW. Regional changes were marginally analyzed in this study. The EEI and its trend are of great importance in climate change, so the topic is relevant and intriguing. The paper is well-written and deserves publication in nature communications. I have several concerns that need to be addressed before publication.

- We would like to thank the reviewer for taking the time and effort to review our manuscript. Each of your comments are addressed below.

Comments:

(1). Page-4, section 1 "Trends in CERES observations". A significant trend in EEI was identified here. However, the CERES data was adjusted by ocean observations for its mean values, only the year-to-year variation is proved to be reliable (see many studies by Dr. Loeb and its group). This adjustment and its impact need to be discussed and assessed. Furthermore, there is a positive trend in recent CERES data, not in previous versions, so I wonder how reliable the trend is for CERES.

- Indeed, our paper is focused on the year-to-year variations and not the absolute mean value of EEI. The reliability of the interannual variations is key for trusting the trend estimate. In fact, Loeb et al., 2018a showed that the CERES EBAF Ed4.0 TOA anomalies are in excellent agreement with a different satellite observational data product: Single Scanner Footprint (SSF1deg) for Terra, Aqua, and National Polar-orbiting Partnership (NPP) satellites. This implies that the 3 different satellites CERES derives its data from, is reliable. Thus, the significant positive trend in EEI is reliable.
- Loeb et al., 2018b showed that lack of a significant positive trend in EEI in CERES EBAF Ed2.8 as compared to CERES EBAF Ed4.0 is due to a smaller reduction in RSW in Ed2.8. This was mostly due to diurnal corrections used in Ed2.8 during the Terra satellite period (March 2000-July 2002) and not due to any calibration differences (Loeb et al., 2018; Su et al., 2020).
- We have summarized this at lines 62-65 and 411-414 in the Introduction and the Methods sections, respectively.

(2). Page-4, section 1 "Trends in CERES observations". As ~90% of EEI was stored in the ocean (AR5), did ocean observations also show a rate increase?

- Yes, according to Cheng et al., 2017 and Von Schuckmann et al., 2020 the ocean heating rate is increasing. They found that the ocean heating rate has been increasing since 1985 and the OHC trend in 1992-2015 was quadruple of 1960-1991 for the upper 700 m of the ocean and increased nine-fold for 700 m-2000 m. In fact, Cheng et al., state "The acceleration is most probably linked to the increasing EEI with time.". Our study indeed shows that EEI is increasing with time. Additionally, Norman Loeb and colleagues are

preparing a manuscript that shows excellent consistency between ocean heating rate's trend and CERES's EEI trend (Loeb, personal communication).

- We have summarized this in the “Trends in CERES observations” subsection of the Results section.

(3). Page-6, line-115. Why 2014 was chosen?

- In our experiments, we wanted to be consistent with where CMIP6-prescribed emissions stop, which is the year 2014. The choice of which year forcing agents are fixed would have no bearing on results since we use interannual anomalies about the mean for computing trends.
- We have summarized this at lines 380-382 in the Methods section.

(4). Figure 1b and associated discussions: why dRSW is negative in all regions? Why dOLR is positive in all latitude regions? More regional analyses are needed to better understand the results. Some geographical maps for model and observation will be appreciated.

- We explain in the section titled “Decomposition of OLR and RSW into forcing and feedback trends” why RSW and OLR trends are negative and positive respectively in all locations. RSW is negative in the tropics because of enhanced East Pacific warming (tropical pattern effect), negative in northern extra-tropics because of an aerosol drawdown, negative in southern extra-tropics because of a Southern Ocean cloud cover decrease (Figure S10b), and negative in the poles because of sea ice melt (Figure S10a). Furthermore, throughout the globe, GHG increases induce cloud cover decreases (Δ ERF component), adding to the negative RSW trend. We have added Supplementary Figures S7, S8, and S10 regarding these findings and includes spatial maps of the trend at each grid point for observations, AM4 PSST+ERF, and AM4 PSST. We refer to these findings in the subsection “Decomposition of OLR and RSW into radiative forcing and response trends”.
- At most latitudes, observations show a decrease in clear-sky OLR due to globe-wide GHG increases, while there is an increase in the longwave cloud radiative effect (LWCRE) (Figure S4, Table S6). This implies that forcing overcame feedbacks in the clear-sky OLR while negative feedbacks in clouds in the LW made the all-sky OLR have a slight positive trend. We have added Supplementary Figures S5 and S15 with spatial maps of the trend at each grid point for observations, AM4 PSST+ERF, and AM4 PSST and mentioned in the subsection “Decomposition of OLR and RSW into radiative forcing and response trends”.

(5). Line-191-192. It seems not uniform to me, at least not for Northern Extra-tropics. Again, some spatial maps were needed to better understand the changes.

- Thank you for pointing that out, indeed, in the northern extra-tropics it remains unchanged. Although we have added the spatial plots in Figure S5 and S15, it is easier to see that the AM4 PSST and AM4 PSST+ERF shading overlap completely in the 30N-60N area, while 30N-70S shows that the AM4 PSST+ERF OLR reduces. We have rephrased this at lines 216-218.

(6). I have some difficulty in understanding “AM4 Fixed Forcing” experiment, why 2014 was chosen, when the EEI has already increased? Why not use the first year 2000 or 2001 as a

baseline? Also the physical meaning of the difference between AM4 Varying Forcing and AM4 Fixed Forcing is also not very clear to me (i.e. Line 189).

- Thank you for letting us know that one of our key experiments is unclear. The AM4 Fixed Forcing experiment (renamed “PSST” now – prescribed SST and sea ice (observed)) is a diagnostic tool that represents the response component of the radiation budget ($\lambda\Delta T_s$). It contains imprints of a forced system, i.e., global warming trend, but does not carry the impact on the radiation budget due to forcing agent changes (solar, volcanic, GHG, aerosol). The AM4 PSST+ERF experiment contains the forcing and response components of the radiation budget ($F + \lambda\Delta T_s$). So, the difference between these two experiments provides an estimate of the radiative forcing changes, i.e., impact of changing forcing agents on the radiation budget. As mentioned earlier, we picked 2014 since this was a convenient year with regard to CMIP6 emissions. It would indeed matter if we cared about the absolute value of EEI which year we picked as the fixed concentration year for the AM4 PSST experiment. However, since we are dealing with trends, if we had used 2000 or 2001 as a baseline, it would not matter. We have summarized this in the subsection “Observed trend in EEI unexplained by internal variability” and lines 373-392 in the Methods section.

(7). Maps for the spatial patterns of the All-forcing, GHG-only forcing, AER-only and NAT-only experiments will be very helpful to illustrate their contributions.

- We have added Supplementary Figures S8 and S15-S16 which contain the spatial patterns of each forcing component for OLR, RSW, and EEI. We mention this in the subsection “Decomposition of OLR and RSW into radiative forcing and response trends”.

(8). Page-9, section “Estimates of forcing and feedback trends, 2001-2018”. There are many convoluted discussions, which needs to be better presented. I suggest to either shorten this section or divide it into two sections.

- Thank you for this suggestion. We have now split up the section into two sections: “Decomposition of EEI into radiative forcing and response trends” and “Decomposition of OLR and RSW into radiative forcing and response trends”. The former section explains how the CERES TEEI is only consistent with the AM4 PSST+ERF experiment and the latter section details how this comes about in its infrared and solar components.

(9). Page-12, lines 221-229. This paragraph is to discuss the contradictions between models and observations. However, it is still not clear how the model uncertainty can impact the main conclusion, which needs an explicit exploration. A comprehensive validation of model simulation (Varying Forcing experiment) against observations can be provided in the supplementary material.

- Thank you for letting us know that it was not clear how model uncertainty can impact the main conclusion. First, our study uniquely provides estimates of model uncertainty due to internal variability (ϵ). Second, when observations and the model do not agree even after accounting for internal variability, model biases could be the cause of the discrepancy. This paragraph discusses the model-observation discrepancies in sea-ice and Southern Ocean cloud cover which cause RSW discrepancies and we accordingly attach estimates. Resolving this discrepancy could have made the model agree exactly with the observed RSW trend and increase the modeled EEI trend to better match the CERES EEI trend.

There are additional uncertainties regarding the exact location of aerosol changes between models and observations – such as RSW over eastern China shows a decrease in the model while observations show an increase. We have summarized this in the Discussion Section.

- We have added a validation of GFDL AM4 against observations in Figure S1. Moreover, the GFDL model has been shown to be able to reproduce observations of climate (Zhao et al., 2018) and one of the highest ranked models in a recent study (Boucher et al., 2020). In their study, GFDL AM4/CM4 is #4/28 in RSW, #3/28 in OLR, #3/26 in SWCRE, #4/26 in LWCRE, and #2/28 in precipitation for CMIP6 alone (ranks are even higher when compared to CMIP5). Furthermore, we have analyzed several more CMIP6 models and our conclusions remain the same (Tables S1-S2). We regard these as a confirmation of the findings obtained here. We have summarized this in subsections “Observed trend in EEI unexplained by internal variability” and “Decomposition of EEI into radiative forcing and response trends”.

Reviewer #2 (Remarks to the Author):

Review of Anthropogenic forcing yields significant positive trend in Earth's energy imbalance by Shiv Priyam Raghuraman, David Paynter , V. Ramaswamy

The manuscript by Raghuraman and colleagues seeks to evaluate the trends in TOA fluxes (EEI, RSW, OLR) using CERES EBAF 4.1 and identify the drivers of the trends using various simulations from GFDL AM4/CM4. The study concludes that anthropogenic forcing plays a key role in observed trends (though it contradicts itself regarding the details) and the regional analysis conducted in the manuscript draws links to the role of sea ice loss (which is known), aerosol trends (which are prescribed), and the tropical pattern effect on changes in albedo (which is also known). The manuscript is fairly well written and the experimental design description is clear. I have major concerns with the interpretation of the simulations and question whether the reasoning and statistical analyses are adequate, and these concerns are detailed below. I also question what is new here, as evident from the comments in parentheses above and suggest that any revised manuscript be explicit regarding findings that are both new and significant. Included are also various minor concerns regarding the lack of important information in the captions/text and the inclusion of excessive detail in others.

- We would like to thank the reviewer for taking the time and effort to review our manuscript. Your comprehensive feedback is very much appreciated and have helped strengthened our manuscript. Each of your comments are addressed in detail below.
- Regarding novelty: we have found a significant signal of acceleration of heat uptake by the planet in the observational record. No other study to the best of our knowledge has shown whether the EEI increase can be attributed to internal variability or forcing and feedbacks in the climate system; we clearly show that it is the latter two components that exert the significant effect, i.e., there is a clear fingerprint of anthropogenic activity on the EEI trend. We have updated our analysis with the addition of nearly fifty CMIP6 models and show that our GFDL model results are robust and that across all models it is exceptionally unlikely the observed TEEI could have been produced by internal variability alone. Some individual mechanisms have been identified before as having an impact on Earth's radiation budget (such as sea-ice loss), however, our hierarchy of modeling experiments convincingly and quantitatively demonstrates that all the individual mechanisms together produce the EEI increase and can be seen in the satellite record. The collective action of the individual mechanisms and the wholesomeness of the climate system interactions in explaining the TEEI is, we believe, a fundamental and original finding.

Major Comments

- There seem to be several internal contradictions in the abstract. For example, the abstract states that the observed trend is only achieved through anthropogenic greenhouse gases. This is odd as Table 1 shows this explicitly not to be the case. Rather aerosols are also fundamental. Moreover, the abstract then goes on to cite aerosol drawdowns as a key part of the TEEI. It then also identifies important contributions from internal variability associated with patterns of tropical warming. Also of this seems inconsistent.

- We have re-written the abstract to clearly explain the TEEI drivers and rectified apparent contradictions as identified by the reviewer.
- The interpretation of Figure 1 seems simplistic as it is framed as trends “with forcing” and “without forcing”. In fact there are vestigial characteristics of forcing in the “without forcing” as this simulation uses observed SST and sea ice. The authors make no effort so far as I can tell to account for these effects and it is unclear if they are fully aware of them. To do this without such contamination one would need fully coupled runs run with/without transient forcing - which would have the effect of increasing the noise and reducing the S/N. Some of the results might not be statistically significant once that is done but I would suggest that is a real hurdle and is what has potentially kept other authors from publishing similar analyses.
 - Your concern regarding the interpretation of the simulations showed that we did not explain the physical significance of the experiments appropriately. We want to be clear here that our aim is to use AM4 to estimate the possible contribution of each of the terms in Equation 1 to TEEI over nearly the last two decades. We have better explained it now earlier in the subsection “Observed trend in EEI unexplained by internal variability”, instead of only in the Methods section and renamed the experiments to make their physical interpretation more apparent. The AM4 Fixed Forcing experiment is now called the prescribed SST and sea ice experiment (PSST). This reflects the fact it has prescribed observed SSTs and sea ice. As the reviewer notes this means that this experiment includes both the response due to unforced variability (ϵ) and feedback to forcing ($\lambda\Delta T_s$), but not the radiative changes from the effective forcing time series (ERF) that leads to that feedback. Hence, this allows us to estimate the contribution of ϵ and $\lambda\Delta T_s$ to TEEI (similar to the technique employed in Gregory and Andrews, 2016). Likewise, the Varying Forcing experiment is now called PSST+ERF to reflect the fact it does contain the effective radiative forcing time-series. The ERF is defined following the RFMIP technique of differencing two runs with and without forcing agents changing (Pincus et al., 2016).
 - The AM4 PSST Experiment was purposely conducted with forcing agents fixed and prescribed observed boundary conditions and we are fully aware the underlying Earth system contained the response of a forced system. Indeed, this experiment provides our estimate of the radiative response ($\lambda\Delta T_s$) in the system already from effective radiative forcing changes. The AM4 PSST+ERF experiment provides the *sum* of ΔERF and $\lambda\Delta T_s$.
 - We would like to clarify that we have not mentioned “without forcing” anywhere in the manuscript. We have used “Fixed Forcing” throughout (renamed now to “PSST” – prescribed SST and sea ice (observed)). We are fully aware of the vestigial characteristics of forcing since we use observed SST and sea ice. Indeed, we account for this and leverage this to calculate $\lambda\Delta T_s$. It is a diagnostic experiment to determine the trends in the system without the radiation responding to forcing agent changes.
 - We provided a climate model with observed SSTs and sea ice boundary conditions so that it can better simulate the *observed* top-of-atmosphere radiation changes. A coupled model with the ERF trend from this 2001-2018 period can reproduce a part of the EEI trend but does not produce the observed La Niñas and El Niños of this period which significantly affect the TOA radiation budget (Figure S12). Thus, coupled model simulations are not representative of all features of the 2001-2018 period. We agree with the reviewer that it is important to use the coupled model without forcing to estimate ϵ

(which we do with an entire suite of CMIP6 models), but to estimate $\lambda\Delta T_s$, we take the approach that is more comparable to observations. The best way to obtain an estimate of how internal variability in the climate system can affect the 2001-2018 EEI trend, is by simulating an initial condition large ensemble with fixed radiative forcing. Other CMIP6 AMIP runs only provide data up to 2014 and almost always only 1 realization. Thus, our AM4 PSST and AM4 PSST+ERF experiments uniquely address the question of what drives CERES's EEI trend. Coupled model 2001-2018 trends in CMIP6 (n=12) do not capture CERES' trend and lie at the lower end of the AM4 PSST+ERF range (Table S5). Future work could explain whether this is because of a weaker ΔERF trend or a stronger $\lambda\Delta T_s$ trend or both. We have added this line of analysis in the Discussion Section.

- Observational uncertainties and particularly the calibration stability uncertainty in CERES has not been adequately accounted for. The challenge is that the design calibration stability of CERES is only slightly less than many of trends being identified. In truth, this uncertainty poses a major challenge for what the authors are trying to do yet it barely gets mention. This is problematic.
 - The Loeb et al., 2018b CERES EBAF data product description does not indicate a systematic error in *trend* arising from observational uncertainties (diurnal corrections, radiance-to-flux conversion, and instrument calibration), so its contribution to our 95% confidence interval uncertainty is neglected. Furthermore, CERES is a stable satellite product, and the CERES team detects and corrects instrument drift (Loeb et al., 2016). For reference, in the ideal case that the regional monthly overall observational uncertainty of 3.5 Wm^{-2} associated with EEI (Loeb et al., 2018) is treated as a random error, then the global monthly observational uncertainty would be 0.01 Wm^{-2} (added in quadrature; $\frac{\sqrt{3.5^2 \times 180 \times 360}}{180 \times 360}$). The addition of such a small uncertainty to each month would cause an even smaller impact on the decadal trend estimate. We have summarized this explanation at lines 416-422. In addition, we would like to mention that in an ideal world we would have 500 years of observational records so as to easily pin down significant trends, instead, we only have a near two-decade record and so we use the next best techniques (95% CI and model ensemble generated internal variability).
 - To make use of all data points provided by CERES, we calculate the trend and 95% CI using monthly anomalies ($0.42 \pm 0.17 \text{ Wm}^{-2}\text{decade}^{-1}$). The trend value using annual anomalies or monthly anomalies is the same. This is consistent with how the CERES team evaluated uncertainty in trends in their data product document, $1.96 \times$ standard error (Loeb et al., 2018b). The March 2000-September 2016 trend in EEI, $0.35 \pm 0.20 \text{ Wm}^{-2}\text{decade}^{-1}$ (95% confidence interval), agrees with the uncertainty value noted in Loeb et al., 2018b, $0.35 \pm 0.24 \text{ Wm}^{-2}\text{decade}^{-1}$. The likely cause of the small $0.04 \text{ Wm}^{-2}\text{decade}^{-1}$ difference in uncertainty is that we use Ed4.1 whereas they use Ed4.0. Other observational studies have also used the standard error as an estimate of observational uncertainty (Hartmann and Ceppi, 2014; Loeb et al., 2016; Chemke and Polvani, 2019; Liu et al., 2020; Su et al., 2020; Chemke and Polvani, 2020). Thus, our uncertainty estimate is consistent with the convention of past studies. We have summarized this in the subsection "Trends in CERES observations".
- Lastly, most of the conclusions in the manuscript are derived from a single model - yet the

authors make almost not case for whether the model is suitable. In this context, there seem to be some clearly disagreement regarding regional model trends and those observed. This should concern the authors. Yet model evaluation doesn't seem to be addressed at all. In my opinion, it needs to be. At minimum, the model's ability to represent CERES mean states and seasonal cycle should be shown to be among the better models given how poorly we know some models to fair in reproducing observed trends. There are also important questions about forcing. The authors state that CMIP6 forcings are used - but don't say which ones. Various groups use different forcings based on the needs of their model (e.g. volcanic). Also, the CMIP6 historical era ends in 2014. What is used after 2014, particularly for aerosols (including biomass burning aerosols)? This is a first-order question in my view but I don't see any discussion.

- We recognize the concern that our study has used only one model. To that end, we have added 47 CMIP6 models in the analysis of internal variability and 6 CMIP6 models in the analysis of radiative forcing trends.
 - The 47 CMIP6 model ensemble has an ϵ of $\pm 0.23 \text{ Wm}^{-2}\text{decade}^{-1}$, in complete agreement with our estimate of $\epsilon \sim 0.2 \text{ Wm}^{-2}\text{decade}^{-1}$. We have listed the values for each model in Table S1.
 - The 6 CMIP6/RFMIP model ensemble has a MMM Net Forcing trend of $0.49 \pm 0.07 \text{ Wm}^{-2}\text{decade}^{-1}$ (95% confidence interval). This is in excellent agreement with our estimate of the radiative forcing trend, $0.51 \pm 0.05 \text{ Wm}^{-2}\text{decade}^{-1}$ (95% confidence interval). We have listed the RFMIP models' forcing trends in Table S2.
 - In addition, we analyzed 12 CMIP6 coupled models in response to your earlier comment. As discussed earlier in Major Comment #2, it will be important future work to determine why the coupled models don't match up to the atmosphere-only simulations and CERES. We have listed the models in Table S7.
- Regarding the GFDL model, we have added Figure S1 in the SI that shows the model's excellent ability to represent the CERES-observed seasonal cycle. Furthermore, Boucher et al., 2020 have shown that the GFDL model is one of the top ranked CMIP6 models in its simulation characteristics of radiation. In their study, GFDL AM4/CM4 is #4/28 in RSW, #3/28 in OLR, #3/26 in SWCRE, #4/26 in LWCRE, and #2/28 in precipitation for CMIP6 alone (ranks are even higher when compared to CMIP5). The excellent simulation by GFDL AM4 of the observed TOA radiation budget allows us to provide physical mechanisms that can explain the observed trends in the system. Regional biases that are found in this paper do not detract from our central findings and their bases.
- We stated in methods that we use SSP2-4.5 after 2014.
- We use the standard CMIP6 forcing datasets (https://docs.google.com/document/d/1pU9IiJvPJwRvIgVaSDdJ4O0Jeorv_2ekEtted34K9cA/edit#heading=h.jdoykiw7tpen). Appendix A of Zhao et al., 2018 provides further details: "Historical reconstructions of monthly solar irradiances are from Matthes et al. (2017). Global monthly mean concentrations of greenhouse gases (GHGs), including carbon dioxide (CO₂), methane (CH₄), and nitrous oxide (N₂O), and ozone depleting substances (ODSs, including CFC-11, CFC-12, CFC-113, and HCFC-22) are from Meinshausen et al. (2017). Annually varying time series of monthly anthropogenic and biomass burning emissions of carbonaceous aerosols and sulfur dioxide (SO₂) precursor to sulfate aerosols, are from the Community Emissions Data System (CEDS; Hoesly et

al., 2017) and the data set of van Marle et al. (2017), respectively.”. Regarding volcanic: “direct injection of SO₂ from volcanic eruptions and emissions of carbonyl sulfide (COS) are not considered in AM4. Instead, we specify time series of stratospheric aerosol optical properties, which includes not only the volcanic contribution to stratospheric aerosol abundance but also other natural and anthropogenic contributions.”. We have now mentioned this in the Methods section.

Line By Line Comments

29: Why do you conclude the driver is anthropogenic greenhouse gases? This is not what the results show.

- We have revised the abstract to appropriately reflect our findings.

42: Sentence begins with a number. Please revise

- Thank you for pointing this out, we have changed it.

64: It is not possible to separate contributions directly from observations but this statement suggests it is. Please revise.

- We have deleted that phrase. Although not directly separable in observations, the model’s separation of the forced component from the unforced component helps understand the drivers of the observational trend.

74: This is not quite right. The simulations you are conducting are AMIP simulations. There are various aspects that are prescribed and thus the question is raised as to what aspects of these are forced and what are internal. There needs to be further discussion around this and I think there are implications for the broader interpretations in the manuscript. For example, if sea ice is prescribed and it is a component of the RSW trend, why do your no-forcing runs not show the associated RSW trend? The same can be argued of any SST forced changes and their associated cloud responses. I thus don’t find the experiments to serve as a suitable contrast between “forced” and “unforced” states, as the authors would like to make them out to be. The discussion of the methods (line 317) seems to recognize this issue with regard to GHG but not other aspects. A complete, comprehensive discussion that addresses these issues and clearly articulates the points being made regarding figure

1 is needed. I am not convinced that a sound argument, making the points the authors seem to want to make, is justified.

- Please see our response to Major Comment #2. We have summarized this in the subsection “Observed trend in EEI unexplained by internal variability” and lines 373-392 in the Methods section.
- We would like to strongly underscore that this paper never contrasts “forced” vs. “unforced” states in the AM4 experiments. We only compare the impact of forcing changes vs. no forcing changes. We are fully aware of the forcing imprints (SST, sea ice trends) in the PSST experiment and utilize this to calculate the response of the climate system during this period. The only difference between the PSST and PSST+ERF experiments are the forcing agents changing in the latter experiment. The difference between the two experiments allows us to compute the impact of changing GHG, aerosol, land-use, etc. on the radiation budget, i.e., effective radiative forcing. The underlying surface conditions are the same.

Fig 1: The authors omit CERES data from 2000, with no justification and with the effect of increasing the magnitude of the estimated trend. They should not. They also should extend the record to be as up to date as possible for the purposes of the discussion.

- We omit 2000 because CERES does not provide January and February data. To compare complete years (Jan-Dec), we begin with 2001. If we include 2000, the EEI trend over March 2000-May 2020 is $0.42 \pm 0.14 \text{ Wm}^{-2}\text{decade}^{-1}$, identical to the value we listed for 2001-2018. Thus, the CERES trend is robust to changing the start and end dates around 2001-2018.
- We have summarized this in the subsection “Trends in CERES observations” and in the Methods section at lines 410-411.

Fig 1: There is no indication on the figure or in the caption what the shading represents. Std err? Std dev? Full range? This should be in the caption. Shouldn't the standard error be shown to make statements regarding the role of forcing and the 2-sigma range used to show the consistency with observations?

- We did indicate in the caption what the shading represents: “Blue shading represents twenty realizations of the AM4 PSST ensemble. Green shading represents twenty realizations of the AM4 PSST+ERF ensemble.”. We have changed it to “...shading represents full range of twenty time series realizations...”.
- We would like to clarify with the reviewer what they mean be standard error should be shown. The aim of this figure is to show the wide range of EEI time series in both experiment ensembles' time series, not trends. The latter is dealt with in Figure 2.

Fig 1: Does not say how regions are defined. Needs to - or state “see text”. But seems to make more sense to put in the figure and remove from text.

- Thanks for the suggestion, we have shifted the regions definitions from the text to the caption in Figure 2.

Fig 1: CERES EBAF4.1 is available into Apr 2020. The data needs to be updated, even if to provide context for the periods of the runs in the discussion.

- Thanks for the suggestion, please see our earlier response to the comment regarding updated CERES trends. We have added in the subsection “Trends in CERES observations” and in the Methods section at lines 410-411.

Fig 1: In order to make statements that are significant, a two std error range needs to be shown and used for the discussion. 1 std error signals are not significant. And why are they being referred to as error bars? And how does one compute standard error from a single member (i.e. obs)? Are these just in regarding to the trend fit? Clearly this is a significant underestimate of the uncertainty in trends.

- We thank the reviewer for pointing out that the one standard error (S.E.) uncertainty we attached to the CERES trend was inconsistent with the model uncertainty attachment of 2σ . Instead, we have now produced two consistent metrics of uncertainty: 1) the CERES trend should be compared to model ensemble's 2σ range (ϵ) and 2) CERES trend with 95% CI should be compared to model ensemble mean and its 95% CI. 95% CI is calculated as $1.96 \times (\text{S.E.})$. In the observational case, the 95% CI is derived from the S.E. associated with the linear fit (Altman and Bland, 2005). In the model ensembles, S.E. is

given by $\frac{\sigma}{\sqrt{n}}$. Either way, our results remain significant, since CERES lies outside the range of control experiments and CERES with 95% CI uncertainty lies above the ensemble mean and 95% CI of the control experiments. Please see subsection “Observed trend in EEI unexplained by internal variability”.

- Yes, the standard error for observations, is the standard error of the linear regression coefficient. This is not a significant underestimate of the uncertainty in the observational trend as mentioned earlier in response to Major Comment #3. We have changed the terminology of “error bar” to “uncertainty given by” and have added explanations at lines 417-422. This two standard error range (95% CI) measures the deviation of the fit from the data points and is broad enough such that it could contain random observational errors.

Fig 1: I find Fig 1b to be redundant with Fig. 2. There is no need for Fig 1b so far as I can tell.

- Thank you for this suggestion, we have deleted Figure 1b. Indeed, it was identical to Figure 2’s left-most panel.

Fig 1: I’m confused by the authors interpretation of Figure 1 and I feel it is not as straight forward as the authors present. Part of the argument is that the climate response in sea ice and SST are what drive EEI. But aren’t those included in AM4 Fixed Forcing? Arguably the lack of greenhouse gas increases the Planck response, the imposed sea ice should reproduce the RSW influence, the SST should drive changes in clouds and water vapor (and thus OLR). And so I don’t think it is fair to interpret the ‘fixed forcing’ as being devoid of the impacts of forcing.

- Again, we would like to thank the reviewer for making us aware that the AM4 PSST experiment design and its physical significance is unclear. Indeed, the climate response in sea ice and SST are included in AM4 PSST.
- We would like to underscore that the PSST experiment certainly is filled with the impacts of forcing; we have not removed greenhouse gases etc. We just held them at fixed concentrations. The lack of greenhouse gas increases and a strong SST trend increases the Planck response and produces a positive trend in OLR. Similarly, the decrease in sea ice causes a negative RSW trend (see earlier response too). We have summarized this in subsection “Observed trend in EEI unexplained by internal variability”.

92: So the key question is what drives the RSW reduction contrast between the runs. It is not some of the key feedbacks, like sea ice, since both are prescribed. If it is forcing, this is pretty unsurprising as the reductions are prescribed. So the key question to me, and the one that needs to be clearly answered to make this a high profile publication, is whether the difference is driven by cloud responses to warming (outside of the Arctic). I don’t think that question has been addressed.

- We would like to thank the reviewer again for making us aware that the AM4 PSST and AM4 PSST+ERF experiment designs, their physical significance, and the difference between the two are unclear. The difference between PSST and PSST+ERF is that *only* the forcing agents are changing, i.e., the atmosphere responds to these forcing agent concentration changes and produces changes in the TOA radiation budget. Thus, the RSW reduction contrast between the runs is driven by aerosols *and* greenhouse gases, amounting to $-0.30 \text{ Wm}^{-2}\text{decade}^{-1}$. First, approximately half of this $-0.30 \text{ Wm}^{-2}\text{decade}^{-1}$ forcing comes from aerosols. The aerosol direct effect may not be

surprising since aerosol reductions are prescribed, but no other study, to the best of our knowledge, has shown this period's observed liquid water path decrease in the northern extra-tropics to be consistent with a model with PSST+ERF. This aerosol-cloud interaction (aerosol indirect effect) significantly contributes to the aerosol forcing component of the all-sky RSW trend and is not prescribed. Second, no study has shown for this period that an equal contributor to the RSW forcing trend comes from the GHG forcing; it makes up the other half of the $-0.30 \text{ Wm}^{-2}\text{decade}^{-1}$, and is also not prescribed. The rapid cloud adjustments have been shown to be important in future global warming simulations before but have not been shown to be a key contributor to the present-day RSW trend. In summary, approximately 60% of the RSW forcing trend comes from clouds (Table S3). We have summarized this in subsection "Decomposition of OLR and RSW into radiative forcing and response trends".

- As the reviewer has mentioned, $-0.12 \text{ Wm}^{-2}\text{decade}^{-1}$ in AM4 PSST comes from the response of the climate system, i.e., prescribed SSTs and sea ice. This $\lambda\Delta T_s$ value largely arises from the tropics due to an enhanced warming in the East Pacific relative to the West Pacific, i.e., tropical pattern effect. This "reverse" pattern effect is a new finding since past studies have shown that the enhanced West Pacific warming caused increases in RSW in the 20th century, while we show the opposite: it decreases RSW in the 21st century. We have summarized this in the subsection "Tropical pattern effect".
- Finally, the model's best estimate of the RSW trend is $-0.42 \text{ Wm}^{-2}\text{decade}^{-1}$. Of which $-0.30 \text{ Wm}^{-2}\text{decade}^{-1}$ and $-0.12 \text{ Wm}^{-2}\text{decade}^{-1}$ come from ΔERF and $\lambda\Delta T_s$, respectively. This modeled $-0.42 \text{ Wm}^{-2}\text{decade}^{-1}$ RSW trend is still $-0.20 \text{ Wm}^{-2}\text{decade}^{-1}$ short of the CERES-observed $-0.62 \text{ Wm}^{-2}\text{decade}^{-1}$ RSW trend. The $-0.20 \text{ Wm}^{-2}\text{decade}^{-1}$ discrepancy largely arises from model-observation sea-ice and Southern Ocean cloud cover discrepancies (Figures S10a,b). This is detailed in the Discussion section.

126: I think you'll find a wide inter-model spread in TEEI variability when you go beyond GFDL models. Need to investigate further such as in the CMIP6 archive.

- We thank the reviewer for suggesting the investigation of internal variability in CMIP6 models. We found that $\epsilon = \pm 0.23 \text{ Wm}^{-2}\text{decade}^{-1}$ across 1,405 CMIP6 realizations (Table S1). This value implies that on average, models' EEI trends can range from $0 \pm 0.23 \text{ Wm}^{-2}\text{decade}^{-1}$ in piControl simulations. This is completely consistent with our estimate of $\epsilon \sim 0.2 \text{ Wm}^{-2}\text{decade}^{-1}$. This robust value of ϵ across all models is encouraging. Furthermore, only 1 realization amongst 1,516 control realizations (CMIP6 models+AM4 Control) has a trend greater than or equal to CERES' 18-year trend. Only 20 realizations have a trend greater than or equal to $0.25 \text{ Wm}^{-2}\text{decade}^{-1}$.

134: But it is almost certainly dependent on model.

- We have deleted this sentence as it detracts from the key results of the paper.

135: Do you find this in the model, the observations, or both? Is it true of models generally?

- We have deleted this sentence as it detracts from the key results of the paper.

Figure 2: It is stated that the error bars on CERES are the standard error? I don't understand how you compute a standard error with 1 member (obs) since standard error should be the standard

deviation divided by the # of members minus one? Perhaps this relates to errors in the trend fit. If it does, I would clarify this further. For the model experiments, why is the standard deviation the relevant metric? Shouldn't the standard error also be shown to make the points that the forced signals are different between the runs (with the std dev used to make the point that CERES observations lie within the model spread).

- We would like to thank the reviewer again for this comment regarding observational uncertainty. Yes, the standard error is the error associated with the slope (trend) from linear regression. Standard error is given by standard deviation divided by # of members (Altman and Bland, 2005). Furthermore, we have now provided 95% confidence intervals for the linear trend ($1.96 \times$ standard error) (Altman and Bland, 2005; Morrison, 2015). As per the reviewer's suggestion, we have shown the 95% confidence interval ($1.96 \times$ standard error = $1.96 \times \frac{\sigma}{\sqrt{n}}$) for the model experiments too. We have added explanations in the Methods section at lines 421-427.

146: Epsilon is not a single value but a $f(t)$ and so it can't be compared to TEEI. I suppose you mean the it is double the largest estimate of the contribution of epsilon to TEEI? The wording needs to be made more precise .

- We have added more details to be more precise in the subsection "Decomposition of EEI into radiative forcing and response trends".

150: Assuming your estimate of internal variability is correct.

- As mentioned earlier, our estimate of internal variability is in excellent agreement with other CMIP6 models ($\epsilon = \pm 0.23 \text{ Wm}^{-2} \text{ decade}^{-1}$; $n=47$).

150: I don't think this is correct. The 0.4 value is the two-sigma range, not the 2-standard error range. Uncertainty in the forced response goes as the 2-standard error range.

- As mentioned in response to Major Comment #3, there are two consistent ways to compare CERES to the model ensembles: 1) CERES trend compared to model 2σ (ϵ) range or 2) CERES trend with 95% CI compared to model ensemble mean and its 95% CI. Please see subsection "Observed trend in EEI unexplained by internal variability".

161: What range is this? Std error or deviation?

- This range was the 2-standard deviation range, now it is the ~ 2 -standard error range ($1.96 \times$ standard error = 95% confidence interval).

164: Again, what range is this? Standard deviation is not the relevant measure.

- This is the 2-standard deviation range, and as mentioned in response to Major Comment #3 and Comment on Line 150 above, this is consistent with the first metric of uncertainty. Please see subsection "Observed trend in EEI unexplained by internal variability".

168: This isn't a prediction.

- We have rephrased this sentence at lines 184-186.

173 and Fig. 2: The trends of the PSST+ERF for the NH extra tropics don't seem to be consistent with the observations for RSW or EEI since the observed value doesn't fall in the ensemble range. Shouldn't this be a cause for concern?

- We thank the reviewer for highlighting this discrepancy. Indeed, EEI in CERES is smaller than AM4 PSST+ERF because the RSW reduction is larger in AM4 PSST+ERF. This is due to larger aerosol and land-use forcing in the model compared to CERES from 45N-60N (Figure S6 b,d,f). In addition, the model shows a reduction in RSW due to aerosol forcing over China whereas observations do not. Observations show reductions in RSW more equatorward than the model (Figure S6b). When the Tropics and NH extra tropics are combined together, the model and observation are in excellent agreement (model and observations each have $-0.39 \text{ Wm}^{-2}\text{decade}^{-1}$).
- We have added this NH extra-tropics model-observation discrepancy in the Discussion section.

There is a lot of language in the figure captions dedicated to explaining the sign of the flux. State it once and be done with it. Rather include more useful information such as what exactly is on the plot.

- We thank the reviewer for this suggestion and have rewritten all captions which now capture the pertinent information.

197: Given this I am perplexed by the claim in the abstract that “this trend is achieved only upon accounting for the increase in radiative forcing by anthropogenic greenhouse gases,”

- We have rewritten the abstract to better capture our findings.

249: Is the gradient in warming the intended index (as stated) or the gradient in SST (as stated in line 248)?

- Since this was confusing and we shortened the caption, this sentence is no longer in the caption.

Fig 3: What is the timescale here? Are these annual mean anomalies? How is the ensemble mean therefore plotted? Do you mean the ensemble mean annual means? This is the kind of information that I’m referring to above in saying more useful information needs to be in the captions.

- We have plotted annual mean anomalies. Yes, the ensemble mean is average of 20 realizations’ annual mean anomalies. We thank the reviewer for this suggestion and have rewritten all captions which now capture the pertinent information.

Fig 3: Why isn’t the mean of the East-West SST difference negative? Surely some removal of the mean has been conducted - yet nothing is mentioned.

- Yes, this is the anomaly in the SST gradient. We have changed the x-axis label to “Anomalies in SST gradient”.

252: What is the uncertainty in the model estimated pattern effect? Surely this is fairly model dependent?

- Our study is uniquely poised to answer how much uncertainty there is in the model estimated pattern effect. Given the same SST pattern, a wide range in tropical RSW is found (AM4 PSST; Figure 3). We quantify the slope and its uncertainty (95% confidence interval) in Table S5, $-0.76 \pm 0.09 \text{ Wm}^{-2}\text{decade}^{-1}$. Since CMIP6 models don’t provide initial-condition large ensembles of the 2001-2018 period with forcing fixed, i.e.,

the equivalent of AM4 PSST, it is hard to quantify what other models produce as the RSW trend in response to the pattern effect. We now specifically say, “this model” in line 283 and mention “See Table S5 for slopes and uncertainty quantification.” in the caption.

Fig 3: Is the point here that the annual means of the CM4 control run exhibit a relationship between the zonal gradient and tropical albedo? And that the control run does as well? Shouldn't the regressions be computed and plotted? and their significance be more rigorously addressed? Is the point in b) that a similar relationship is expressed in trends in albedo and SST gradient? Again a significant amount of text in the caption is dedicated to describing simple sign conventions and not converting important material about the figure such as these aspects.

- We thank the reviewer for these clarification questions. Yes, the point here is that CM4 Control, AM4 PSST, and CERES (without forcing changes) exhibit a negative relationship between the zonal gradient and tropical albedo. We have computed the regressions along with their 95% confidence intervals to show statistical significance in Table S5. We have rewritten all captions which now capture the pertinent information.

277: “estimate” present tense.

- Thank you, we have corrected this.

308: There is no discussion of drift in the coupled control. Is it zero? What is the TOA imbalance?

- We plot the CM4 Control EEI time series in Figure S13a-c. This figure and the related analyses exclude the first 18 years due to drift. The trend through this period is negligible, $-0.002 \pm 0.007 \text{ Wm}^{-2}\text{century}^{-1}$, implying that it can be used for the analysis. The mean EEI (TOA imbalance) is 0.28 Wm^{-2} . We have added this at lines 361-364 in the Methods section.

311: Are the ESM2M or CM3 used anywhere in the manuscript? I see there is mention of trends in the PI Control - but really a more comprehensive estimate from CMIP6 should be used if the goal here is to show model robustness. These data are freely available.

- We thank the reviewer again for this suggestion of adding CMIP6 model estimates of internal variability. ESM2M and CM3 are only used for the estimate of internal variability from their piControl runs. Indeed, we show that our GFDL model results are robust since our estimate of $\epsilon \sim 0.2 \text{ Wm}^{-2}\text{decade}^{-1}$ is consistent with the CMIP6 $\epsilon = \pm 0.23 \text{ Wm}^{-2}\text{decade}^{-1}$ (n=47). We have added the CMIP6 estimates in Table S1 and a methods description regarding CMIP6 ϵ at lines 368-373 in the Methods section.

314: Would be useful to state what datasets are used for the CMIP6 AMIP runs rather than reference other manuscripts.

- Please see our response to Major Comment #4 for the datasets.

319: The upshot is that your PSST runs have several contrasting trends that are difficult to interpret. Some are associated with the forced response in imposed observations (SST, sea ice and associated changes) and some associated with internal variability. This makes the runs difficult to interpret.

- We would again like to thank the reviewer for making us aware that the experiment design and its physical significance was unclear. Our AM4 PSST experiment was designed to and did separate the trend due to the forced response (ensemble mean) from the internal variability (2σ range of trends). For example, the AM4 PSST global EEI trend of $-0.22 \text{ Wm}^{-2}\text{decade}^{-1}$, represents the trend due to the forced response inherent in the imposed observations (SSTs and sea ice). The 2σ spread associated with the AM4 PSST EEI trend of $\pm 0.19 \text{ Wm}^{-2}\text{decade}^{-1}$ is due to internal variability in the climate system. Since we want to compare to observations, AMIP-style runs are the best possible way to analyze changes in the TOA radiation budget.

343: A statement regarding the estimated calibration stability of CERES is warranted in addition to a statement regarding the consequences for the results and its basis.

- We thank the reviewer again for expressing their concern related to observational uncertainties. As detailed in the response to Major Comment #3, observational uncertainties in decadal trends due to systematic errors make negligible contributions to the existing 95% confidence interval uncertainty. We have added this in subsection “Trends in CERES observations” and at lines 415-423 in the Methods section.

347: Standard errors of the linear fit are only a subset of the uncertainties. Should the calibration stability uncertainties in CERES also be considered?

- As with the above response and earlier responses, calibration stability uncertainty (one of the three main observational uncertainties) exists but does not affect our trend estimates unless they are systematic errors. We are unaware of systematic errors that could skew the CERES trends in a particular direction; if the reviewer has a specific source regarding systematic error, we can include it. We have added this at lines 415-423 in the Methods section.

Table 1: This is a decomposition of the contribution to the trends, not the absolute forcing. This should be clarified.

- Thank you for pointing this out, we have changed it to trend.

The effective radiate forcing trend is only slightly larger than the change in EEI. This seems odd given that the present day imbalance is considerably less than the forcing (~50%). Why is this?

- The reviewer raises a great question. Qualitatively, this could be due to the fact that this period is more sensitive (small $\lambda\Delta T_s$) relative to the historical period. It also could be because over a long period of time, the depths of the ocean adjust to imposed radiative forcing and have a stronger $\lambda\Delta T_s$ while on a short time period such as the last two decades, the ocean has not had the time due to thermal inertia. However, a rigorous quantitative analysis of this topic should be done in future studies.

Reviewer #3 (Remarks to the Author):

“Anthropogenic forcing yields significant positive trend in Earth’s energy imbalance” by Raghuraman et al.

The content of the paper is interesting and useful for the EEI study, since the authors have tried to find the contributing drivers for the TEEI. The idea is quite novel and the method should be straightforward. However, I found it difficult to follow because of the vague description of the method details. I suggest to return to the authors for further revision.

- We would like to thank the reviewer for taking the time and effort to review our manuscript. Each of your comments are addressed below.

Major comments

As I understand, in AM4 control, the years are randomly sampled to form 18-year time series, and in CM4 control run, the consecutive 18 year period is sampled by randomly selecting the start year.

- We thank the reviewer for letting us know that the experiment design description was not clear. Yes, in AM4 Control, years from a 200-year time series are randomly sampled to form a 2000-year time series (Figure S13d). Next, 111 18-year periods (1998 years totally; excludes last 2 years) are sampled consecutively. These 111 18-year periods provide 111 linear trends. The 2σ spread of these 111 trends is our estimate of ϵ for AM4 Control.
- In CM4 Control (piControl run), 18-year periods are sampled consecutively: 0-17, 18-35,...then the linear trend is calculated for each period. This creates 35 18-year period trends. The 2σ spread of these 35 trends is our estimate of ϵ for CM4 Control. We have added this at lines 350-372 in the Methods section.

Do you get ϵ by fitting Eq (1)?

- The trend for an 18-year ΔEEI (or RSW or OLR) time series is the linear fit/slope through the 18-year period. We calculate this trend for each 18-year period in each model ensemble: AM4 Control has 111 18-year periods, CM4 Control has 35 18-year periods, AM4 PSST has 20 18-year periods, and AM4 PSST+ERF has 20 18-year periods. The 2σ spread of these 111, 35, 20, and 20 trends for each ensemble experiment provides our estimate of ϵ (units: $\text{Wm}^{-2}\text{decade}^{-1}$). ϵ is the range of trends in EEI or RSW or OLR due to internal variability. We have added this at lines 350-372 in the Methods section.

It is not clear how ΔEEI is calculated. I can guess ΔT s is calculated from monthly anomaly, but not ΔEEI .

- ΔEEI represents interannual anomalies and is calculated by removing the mean EEI of the period, from each year. For example, $\Delta\text{EEI}_{2018} = \text{EEI}_{2018} - \overline{\text{EEI}_{2001-2018}}$. We have added this at lines 344-347 in the Methods section.

Or the ΔEEI trend in the control run is regarded as the ϵ trend?

- We would like to thank the reviewer again for letting us know that the method in which ϵ was calculated was unclear. As described in the responses to the first two major comments, ϵ is the 2σ range of trends in each model ensemble. Thus, CM4 Control’s

estimate of ϵ is 1 of the 4 estimates of ϵ . A key result of this manuscript is that the 4 different ensembles' estimates of ϵ have nearly the same $\epsilon \sim 0.2 \text{ Wm}^{-2} \text{ decade}^{-1}$.

Minor comments

Lines 32-33: “The decrease in reflected solar radiation is due to midlatitude aerosol drawdown, spatial pattern changes in tropical warming, and polar sea-ice decrease” This is well known fact, not the new finding.

- Thank you for this comment. Our work found a significant signal of acceleration of heat uptake by the planet in the observational record. No other study to the best of our knowledge has shown whether the EEI increase can be attributed to internal variability or forcing and feedbacks in the climate system; we clearly show that it is the latter two components that exert the significant effect, i.e., there is a clear fingerprint of anthropogenic activity on the EEI trend. The collective action of the individual mechanisms and the wholesomeness of the climate system interactions in explaining the TEEI is, we believe, a fundamental and original finding. We have revised the abstract to appropriately reflect these new findings.

Line 48: It is not clear how the “interannual anomaly” is defined?

- As explained in the earlier response to the Major Comment, ΔEEI represents interannual anomalies and is calculated by removing the mean EEI of the period, from each year. For example, $\Delta\text{EEI}_{2018} = \text{EEI}_{2018} - \overline{\text{EEI}_{2001-2018}}$. We have added this at lines 344-347 in the Methods section.

Line 50 : “can expressed” should be “can be expressed”

- Thank you for pointing this out, we have changed this.

Is ΔEEI the EEI monthly anomaly?

- EEI is provided in monthly values but we average it into annual values. As explained in the earlier response to the major comment, ΔEEI represents interannual anomalies. Only for CERES, to make use of all months available, we calculate the trend through the monthly anomaly time series. The trend using monthly or annual anomaly time series is the same. We have added this at lines 344-349 in the Methods section.

Line 73: “This provides a 2σ range of trends as our estimate of”, what does this mean? How do you calculate it? Please give details in the method.

- We thank the reviewer again for letting us know that our explanation of how ϵ is calculated was unclear. We have added this at lines 350-372 in the Methods section.

Figure 1a:

It would be better to show fitted lines for other two cases as well.

It is not clear how you get the error bar? Do you get the fitted slopes first, and then get the mean and STD of the slopes next?

- Thank you for this suggestion. We have added the trend lines for the ensemble mean ΔEEI time series for both experiments.

- In Figure 1a, the shading is the full range of time series in each ensemble. The error bar in Figure 1b is standard error associated with slope of linear fit. Please note that Figure 1b is now encompassed in the new Figure 2.
- Furthermore, all error bars in Figure S2 (complementary to Figure 2) represent 95% confidence intervals. For observations, the error bar is given by $1.96 \times$ (standard error associated with slope of linear fit). For each model ensemble, the method is as the reviewer has summarized: the standard deviation (1σ spread) of the ensemble's trends (slope/linear fit) is used to calculate the 95% confidence interval: $1.96 \times \frac{\sigma}{\sqrt{n}}$, where n =number of ensemble members.

Lines 120-121: "We estimate ϵ by computing the 2σ spread of trends in each ensemble (Figure 2)." It is not clear how ϵ is estimated.

- We thank the reviewer again for letting us know that our explanation of how ϵ is calculated was unclear. We have added more details at lines 79-82, 116-122, 129-130, 137-138, and 350-372.

References used in above responses to referees

1. Loeb, Norman G., et al. "Changes in earth's energy budget during and after the "pause" in global warming: an observational perspective." *Climate* 6.3 (2018): 62.
2. Loeb, Norman G., et al. "Clouds and the earth's radiant energy system (CERES) energy balanced and filled (EBAF) top-of-atmosphere (TOA) edition-4.0 data product." *Journal of Climate* 31.2 (2018): 895-918.
3. Cheng, Lijing, et al. "Improved estimates of ocean heat content from 1960 to 2015." *Science Advances* 3.3 (2017): e1601545
4. Von Schuckmann, Karina, et al. "Heat stored in the Earth system: where does the energy go?." *Earth System Science Data* 12.3 (2020): 2013-2041.
5. Zhao, Ming, et al. "The GFDL global atmosphere and land model AM4. 0/LM4. 0: 1. Simulation characteristics with prescribed SSTs." *Journal of Advances in Modeling Earth Systems* 10.3 (2018): 691-734.
6. Boucher, Olivier, et al. "Presentation and evaluation of the IPSL-CM6A-LR climate model." *Journal of Advances in Modeling Earth Systems* 12.7 (2020): e2019MS002010.
7. Altman, Douglas G., and J. Martin Bland. "Standard deviations and standard errors." *Bmj* 331.7521 (2005): 903.
8. Hartmann, Dennis L., and Paulo Ceppi. "Trends in the CERES dataset, 2000–13: The effects of sea ice and jet shifts and comparison to climate models." *Journal of climate* 27.6 (2014): 2444-2456.
9. Loeb, Norman G., et al. "CERES top-of-atmosphere earth radiation budget climate data record: Accounting for in-orbit changes in instrument calibration." *Remote Sensing* 8.3 (2016): 182.
10. Chemke, Rei, and Lorenzo M. Polvani. "Opposite tropical circulation trends in climate models and in reanalyses." *Nature Geoscience* 12.7 (2019): 528-532.
11. Liu, Chunlei, et al. "Variability in the global energy budget and transports 1985–2017." *Climate Dynamics* 55.11 (2020): 3381-3396.

12. Su, Wenying, et al. "Uncertainties in CERES Top-of-Atmosphere Fluxes Caused by Changes in Accompanying Imager." *Remote Sensing* 12.12 (2020): 2040.
13. Chemke, R., and L. M. Polvani. "Using multiple large ensembles to elucidate the discrepancy between the 1979–2019 modeled and observed Antarctic sea ice trends." *Geophysical Research Letters* 47.15 (2020): e2020GL088339.
14. Morrison, Faith A. "Obtaining uncertainty measures on slope and intercept of a least squares fit with Excel's LINEST." *Houghton, MI: Department of Chemical Engineering, Michigan Technological University. Retrieved August 6 (2014): 2015.*
15. Gregory, Jonathan M., and T. Andrews. "Variation in climate sensitivity and feedback parameters during the historical period." *Geophysical Research Letters* 43.8 (2016): 3911-3920.
16. Pincus, Robert, Piers M. Forster, and Bjorn Stevens. "The radiative forcing model intercomparison project (RFMIP): Experimental protocol for CMIP6." *Geoscientific Model Development (Online)* 9.9 (2016).

REVIEWER COMMENTS

Reviewer #1 (Remarks to the Author):

I think the authors have addressed all my concerns and other reviewers' comments, the paper is significantly improved and ready to go.

Reviewer #2 (Remarks to the Author):

Anthropogenic forcing and response yield positive trend in Earth's energy imbalance
Raghuraman et al.

Recommendation: Reject.

First, let me apologize for the lateness of my review as it has taken a few days longer than I originally expected. As I understand it, the goal of this work is to show that the positive trend in Earth's energy imbalance is a response to anthropogenic forcing during the CERES era. While this is an unsurprising result (as it is well-known to be the driver of the long-term trend), it is a challenging task because of the inherent noise in the climate system arising from internal variability and the brevity of the CERES record. For some reason the authors have chosen to make it more challenging still by shortening the CERES record from the fully available record. They provide a compelling case that, at least in an AMIP configuration, trends in the forcing agents are needed to achieve consistency with observed trends in the model.

That said, I continue to have reservations about the analysis, foremost being the use of AMIP runs to address what is inherently a coupled problem. The survey of key literature and use of available methods for tackling this type of problem also go largely unaddressed. At its core this is a detection and attribution study. There is a broad body of D&A work that has been done over the past decade that this work seems entirely unaware of as indicated by its failure to discuss any of the relevant D&A methods (though it does cite some of the relevant manuscripts). I refer the authors to the relevant IPCC AR5 chapter <https://www.ipcc.ch/report/ar5/wg1/detection-and-attribution-of-climate-change-from-global-to-regional/>. Instead this work uses a new and largely untested approach based on AMIP simulations. Much of the discussion relies on imprecise terminology regarding "internal variability", as evidenced by its statement in line 29 that the observed trend is unlikely to be explained by "internal variability". So far as I can tell, this has only been demonstrated for atmospheric internal variability with the assumption that if one prescribes SSTs in a simulation then the ocean component can be ignored. But can it? Why not simply examine a fully coupled large ensemble with full forcing and a similar large ensemble with fixed forcings (at 2000 levels?) and compare their trends during the CERES era? Such an analysis would be considerably more convincing, though I suspect it may result in a null result.

There are also gaps in the manuscript as some of the central questions, at least in my view, go poorly answered or unaddressed altogether. Some statements are just wrong, such as what is

actually shown by Held et al. regarding GFDL-CM4. For example the statement in the introduction claiming that the fidelity of GFDL/CM4 is superior to ‘nearly every CMIP6 class model when compared to observations’ - this is simply wrong as Held et al. don’t even look at CMIP6 models. Instead they present comparisons to CMIP5 models - such as Fig 15 - (in part likely out of necessity as many CMIP6 simulations were unavailable at the time of publication). As suggested, there is also no need to end CERES data in 2018 when in fact the longer record would benefit any S/N analysis. A central assertion in this work is that “TEEI is achieved only upon accounting for the increase in anthropogenic radiative forcing and the associated climate response”, which I continue to question because the statement relies on the fidelity of the internal variability of the model - and the methods used to estimate the coupled variability do not account for the influence of climate change on it. Lastly, while SST may seem to be a well-observed fields there are several published works that show that the SST dataset used can have a large impact on deep convection in the warm pool and simulated climate trends in AMIP configurations. What is the sensitivity of these results to the choice of SST dataset? The reader is left to wonder.

For these reasons, I find myself having more questions than answers after reading the manuscript and I do not find it suitable for publication in its present form. A cleaner analysis, and perhaps one that uses standard D&A methods, would in my view be more convincing and represent a nice contribution to the literature. I do look forward to reading such a contribution.

Detailed points:

Please update your data through the end of the CERES record. It is available though July 2020.

65: There should be some discussion on the reliability of the trends and what the error ranges are since TEEI is the central focus of the manuscript. There are estimates of instrument calibration but these are swept aside.

68: This work also does not separate the forced and unforced components.

76: This statement both references the wrong manuscript in the reference list and provides an inaccurate summary of Held et al.

80: The statements made really should clarify that these are trends in AMIP mode - not in coupled mode.

94: One simple but sometimes useful way to distinguish between forced and internal variability is to examine the seasonality of a signal. This would seem to be an important step to consider to bolster the conclusions given the other concerns mentioned.

100: Is this surprising?

110: This discussion and treatment of CERES uncertainty and error is unjustified.

Note the reference to GFDL's summary manuscript is wrong (should be 31) and the statement that it performs better than nearly every CMIP6 class model is similarly wrong - Held et al. compare to CMIP5 models.

The GFDL-CM4 has a huge 20th C "pothole" see Held et al. This should give one pause when using it for trends in the CERES era - this is a major issue.

123: This seems to assume that internal variability is assumed to be unchanged by climate change. In fact it is not, particularly in the GFDL model.

131: This epsilon only includes unforced atmospheric variability?

146: In none of your experiments do you estimate coupled variability during the CERES era. I view this to be a major issue given that this is a S/N exercise.

147: Again - epsilon needs to be more precisely discussed. Distinctions need to be made regarding estimates from AMIP PI, AMIP PD, coupled PI, and coupled PD

477: Have you explored the sensitivity of the GFDL model to other SST datasets?

<https://agupubs.onlinelibrary.wiley.com/doi/abs/10.1002/2014JD022365>

Responses to Reviewers' Comments on "Anthropogenic forcing and response yield positive trend in Earth's energy imbalance"

Reviewer #1 (Remarks to the Author):

I think the authors have addressed all my concerns and other reviewers' comments, the paper is significantly improved and ready to go.

Thank you. We appreciated your suggestions; they helped significantly improve the manuscript.

Reviewer #2 (Remarks to the Author):

Anthropogenic forcing and response yield positive trend in Earth's energy imbalance
Raghuraman et al.

Recommendation: Reject.

We thank the reviewer for taking the time to review our manuscript. We appreciate your suggestions. We have responded to each of your concerns below.

First, let me apologize for the lateness of my review as it has taken a few days longer than I originally expected. As I understand it, the goal of this work is to show that the positive trend in Earth's energy imbalance is a response to anthropogenic forcing during the CERES era. While this is an unsurprising result (as it is well-known to be the driver of the long-term trend), it is a challenging task because of the inherent noise in the climate system arising from internal variability and the brevity of the CERES record.

The state of the science, as mentioned in the Intergovernmental Panel on Climate Change Sixth Assessment Report Chapter 7 Second Order Draft, still assumes that the trend in EEI is driven by internal variability (page 7-15, line 27). This is further reflected in the recent paper by the CERES EBAF team (Loeb et al., 2018). Our work shows that the observed trend is too large to be consistent with internal variability seen in both coupled model and AMIP experiments. In the plot below (Figure R1 and Figure S15) we show that the observed trend was still in the envelope of internal variability for the first 15 years of the satellite record despite the forcing increasing. We have elaborated more on this in the Discussion section.

For some reason the authors have chosen to make it more challenging still by shortening the CERES record from the fully available record.

We have extended our simulations and results till December 2020, the latest month of CERES EBAF Ed4.1 data. Thank you for your suggestion.

They provide a compelling case that, at least in an AMIP configuration, trends in the forcing agents are needed to achieve consistency with observed trends in the model.

We have now included more CMIP6 coupled models (historical) in our analysis to back up our claims about our AMIP results. Although we would like to note that the previous version of the paper did include 47 CMIP6 coupled model (CMIP6 Control) estimates of internal variability, we acknowledge that we perhaps did a poor job of highlighting these results. As you will see below and in the main text, we have prominently displayed the coupled model results by adding two new figures (Figure 2 and Figure 4).

Figure R1 | EEI trend vs trend length for CERES EBAF satellite data and GFDL CM4 Control. Shading indicates 2σ range of trends, i.e., ϵ as referred in the main text.

That said, I continue to have reservations about the analysis, foremost being the use of AMIP runs to address what is inherently a coupled problem. The survey of key literature and use of available methods for tackling this type of problem also go largely unaddressed. At its core this is a detection and attribution study. There is a broad body of D&A work that has been done over the past decade that this work seems entirely unaware of as indicated by its failure to discuss any of the relevant D&A methods (though it does cite some of the relevant manuscripts). I refer the authors to the relevant IPCC AR5 chapter <https://www.ipcc.ch/report/ar5/wg1/detection-and-attribution-of-climate-change-from-global-to-regional/>. Instead this work uses a new and largely untested approach based on AMIP simulations. Much of the discussion relies on imprecise terminology regarding “internal variability”, as evidenced by its statement in line 29 that the observed trend is unlikely to be explained by “internal variability”. So far as I can tell, this has only been demonstrated for atmospheric internal variability with the assumption that if one prescribes SSTs in a simulation then the ocean component can be ignored. But can it? Why not simply examine a fully coupled large ensemble with full forcing and a similar large ensemble with fixed forcings (at 2000 levels?) and compare their trends during the CERES era? Such an analysis would be considerably more convincing, though I suspect it may result in a null result.

Thank you for your suggestion regarding determining epsilon in fully coupled models with forcing. We have now added 5 CMIP6 large ensembles (LEs; 142 realizations totally) with

historical forcing till 2014 and SSP2-4.5 forcing after 2014 to our analysis. We analyzed all CMIP6 models available with at least 10 realizations per ensemble. We find that the epsilon in the coupled model forcing LEs during 2001-2020 is the same as the epsilon of the coupled models with forcing fixed at 1850 (piControl). Please see Figure 2 in the main text; copied below as Figure R2 for your convenience.

As noted above, we would like to remind the reviewer that our estimate of epsilon in our last submission (January 11) not only came from the GFDL AM4 AMIP simulations, but also came from 47 CMIP6 fully coupled models' piControl runs. We had demonstrated that the observed trend is extremely unlikely to be explained by internal variability based on the robust estimate of epsilon across models, fully coupled and atmosphere-only. In fact, the statement on line 29 and its associated probability was primarily based on the 1,405 realizations of the 47 CMIP6 fully coupled piControl runs.

We thank the reviewer again for the suggestion of looking at CMIP6 coupled models with forcing. We find that almost all coupled models (exception being MIROC6) with forcing fall into the 95% confidence interval of CERES (CanESM5 splits into 'p1' and 'p2' physics configurations and each fall into the CERES 95% CI). This implies that anthropogenic forcing and response indeed cause the TEEI in coupled models as well. Please see Figure 4 in the main text; copied below as Figure R3 for your convenience. On a related note, we disagree that at the core this is a detection and attribution study. D&A in IPCC AR5 relates primarily to identifying human influences on temperature and surface variables using spatial fingerprinting techniques. This strict IPCC definition of D&A is not done for radiation changes. Only loosely can our work be construed as D&A because it detects changes in Earth's radiation balance, and we identify its causal factors. Our work is more along the lines of past studies that identified human influences on various atmospheric and oceanic phenomenon (e.g., Vecchi et al., 2006; Bollasina et al., 2011; Cai et al., 2014).

Figure R2| Global-mean estimates of maximum trends in EEI due to internal variability (ε).

Figure R3| Trends in EEI with 95% confidence intervals. Shading indicates CERES TEEI $\pm \frac{\epsilon}{\sqrt{1}}$ as the 95% CI. The lower end indicates the minimum contribution by anthropogenic forcing and the associated climate response ($\Delta\text{ERF} + \lambda\Delta T_s$) to CERES TEEI. Conversely, the upper end indicates the maximum contribution by $\Delta\text{ERF} + \lambda\Delta T_s$ to CERES TEEI. Dashed lines indicate CERES TEEI 95% CI derived from standard error of linear fit. GFDL AM4 PSST+ERF (AMIP) represents the ‘Historical’ value plotted.

There are also gaps in the manuscript as some of the central questions, at least in my view, go poorly answered or unaddressed altogether. Some statements are just wrong, such as what is actually shown by Held et al. regarding GFDL-CM4. For example the statement in the introduction claiming that the fidelity of GFDL/CM4 is superior to ‘nearly every CMIP6 class model when compared to observations’ - this is simply wrong as Held et al. don’t even look at CMIP6 models. Instead they present comparisons to CMIP5 models - such as Fig 15 - (in part likely out of necessity as many CMIP6 simulations were unavailable at the time of publication). As suggested, there is also no need to end CERES data in 2018 when in fact the longer record would benefit any S/N analysis. A central assertion in this work is that “TEEI is achieved only upon accounting for the increase in anthropogenic radiative forcing and the associated climate response”, which I continue to question because the statement relies on the fidelity of the internal variability of the model - and the methods used to estimate the coupled variability do not account for the influence of climate change on it. Lastly, while SST may seem to be a well-observed fields there are several published works that show that the SST dataset used can have a large impact on deep convection in the warm pool and simulated climate trends in AMIP configurations. What is the sensitivity of these results to the choice of SST dataset? The reader is left to wonder.

We think that the reviewer might have accidentally linked the wrong reference to the statement in the text. We say this because we do not reference Held et al., 2019 (reference 31) in reference to the statement regarding GFDL CM4/AM4’s fidelity. We refer to Boucher et al., 2020 (reference 32) for that. The image below (Figure R4) is a screenshot of the January 11 submitted

manuscript on the Nature Communications manuscript tracking system showing that reference 32 refers to Boucher et al., 2020 and not Held et al., 2019. We have rephrased this sentence in the introduction. However, we stand by our previous response to the reviewer from the last round (January 11 “Responses to Reviewers’ Comments” document on page 7). We copy this again below: “Boucher et al., 2020 have shown that the GFDL model is one of the top ranked CMIP6 models in its simulation characteristics of radiation. In their study, GFDL AM4/CM4 is #4/28 in RSW, #3/28 in OLR, #3/26 in SWCRE, #4/26 in LWCRE, and #2/28 in precipitation for CMIP6 alone (ranks are even higher when compared to CMIP5).”

As noted above, we have extended the simulations and results through to December 2020. Regarding the signal to noise analysis and coupled models: we find that during this 2001-2020 period, 3 of the 5 CMIP6 fully coupled model large ensembles with forcing (CMIP6 Historical) show the signal to noise ratio to be greater than 2. The remaining 2 models cross the SNR=2 in just a couple of years, 2022 and 2023 (Figure R5 below). Overall, this implies that the forced signal indeed overcomes noise over this period or shortly thereafter in fully coupled models with forcing.

We believe that our assertion is valid since it relies on not just one model’s internal variability but relies on 47 fully coupled models with fixed forcing (CMIP6 Control), 5 fully coupled models with varying forcing (CMIP6 Historical; only models with 10 or more realizations were considered so that they form large ensembles), 1 atmosphere model with fixed forcing and fixed SSTs (AM4 Control), 1 atmosphere model with fixed forcing but prescribed SSTs and sea ice (AM4 PSST), and 1 atmosphere model with varying forcing and prescribed SSTs and sea ice (AM4 PSST+ERF). We thank the reviewer for the suggestion of estimating the coupled variability with the influence of climate change on it (CMIP6 Historical); indeed, the estimate remains unchanged (Figure R2).

We acknowledge the reviewer’s point that the SST dataset influences atmospheric temperature trends. However, exploring the sensitivity of SST datasets to the radiation budget changes is out of the scope for this paper. Future work could explore this exciting possibility. Indeed, some work has already been done on this (e.g., Andrews et al., 2015).

Figure R4 | January 11 submitted manuscript on the Nature Communications manuscript tracking system.

Figure R5 | Signal-to-noise ratio (SNR) for CMIP6 Historical large ensembles. $SNR = \frac{\overline{Trends}}{\sigma_{Trends}}$, where the overbar denotes the mean and σ denotes the standard deviation of the trends in each model large ensemble. Note that 3/5 of the models emerge above internal variability

(SNR ≥ 2) during 2001-2020. The other 2 models (MIROC6 and ACCESS-ESM1.5) emerge shortly after in 2022-2023.

For these reasons, I find myself having more questions than answers after reading the manuscript and I do not find it suitable for publication in its present form. A cleaner analysis, and perhaps one that uses standard D&A methods, would in my view be more convincing and represent a nice contribution to the literature. I do look forward to reading such a contribution.

We thank the reviewer for raising their concerns. As detailed above and further detailed below, our revisions display a cleaner analysis.

Detailed points:

Please update your data through the end of the CERES record. It is available through July 2020.

Thank you for your suggestion, we have updated our simulations and results through December 2020.

65: There should be some discussion on the reliability of the trends and what the error ranges are since TEEI is the central focus of the manuscript. There are estimates of instrument calibration but these are swept aside.

We would like to remind the reviewer that we did discuss the reliability of the trends, CERES data, and error ranges in the January 11 submission's main text at lines 108-113, 143-168, and 417-429. This submission's equivalent lines are: 102-108, 157-189, 427-437.

68: This work also does not separate the forced and unforced components.

We have rephrased this sentence.

76: This statement both references the wrong manuscript in the reference list and provides an inaccurate summary of Held et al.

Please see the earlier response above for more details (Figure R4). We have revised this statement.

80: The statements made really should clarify that these are trends in AMIP mode - not in coupled mode.

We have made this more specific; here we are referring to the estimate of epsilon from CMIP6 fully coupled model piControl simulations.

94: One simple but sometimes useful way to distinguish between forced and internal variability is to examine the seasonality of a signal. This would seem to be an important step to consider to bolster the conclusions given the other concerns mentioned.

In Figure S1 we show the seasonal cycle and the model ensemble's range. Future research could further investigate the seasonal trends in models and observations.

100: Is this surprising?

The fact that every latitude has an increase in EEI is worth noting in our opinion.

110: This discussion and treatment of CERES uncertainty and error is unjustified.

Please see our response for line 65 above. We justified this in the last round of revisions (January 11 submission).

Note the reference to GFDL's summary manuscript is wrong (should be 31) and the statement that it performs better than nearly every CMIP6 class model is similarly wrong - Held et al. compare to CMIP5 models.

Please see our response for line 76 and the response accompanying Figure R4 above. Held et al., 2019 refers to 31 and the statement draws on Boucher et al., 2020 (reference 32).

The GFDL-CM4 has a huge 20th C "pothole" see Held et al. This should give one pause when using it for trends in the CERES era - this is a major issue.

Thank you for pointing out this issue. Many CMIP6 models display this issue but can still be used to draw conclusions regarding radiative trends in the CERES era. Coupled model historical runs have the same estimate of internal variability as AMIP runs. Furthermore, since the AMIP runs use prescribed observed SSTs and sea ice, they reproduce the top-of-atmosphere radiation budget changes well as evidenced by Figure 2. We are therefore not completely reliant on just coupled model simulations (nor are we just reliant on AMIP simulations).

123: This seems to assume that internal variability is assumed to be unchanged by climate change. In fact it is not, particularly in the GFDL model.

Please see our responses for the major comment accompanying Figure R2 above. We show that the estimate of internal variability remains unchanged with or without forcing changes in fully coupled models in PI and PD.

131: This epsilon only includes unforced atmospheric variability?

Yes, the AM4 PSST experiment's epsilon includes only unforced atmospheric variability as well as land temperature variability. This experiment has prescribed SSTs and sea ice and fixed forcing agents.

146: In none of your experiments do you estimate coupled variability during the CERES era. I view this to be a major issue given that this is a S/N exercise.

Please see our responses for the major comment accompanying Figure R2 above. We thank the reviewer again for this suggestion of estimating the coupled variability during the CERES era.

147: Again - epsilon needs to be more precisely discussed. Distinctions need to be made regarding estimates from AMIP PI, AMIP PD, coupled PI, and coupled PD

We apologize for the lack of clarity of which epsilon is being discussed. We have made this more precise in the results section. We have 5 estimates of epsilon: AM4 Control (AM PI), AM4 PSST (AMIP PD), AM4 PSST+ERF (AMIP PD), CMIP6 piControl (coupled PI), CMIP6 Historical (coupled PD). All of them are nearly identical.

477: Have you explored the sensitivity of the GFDL model to other SST datasets? <https://agupubs.onlinelibrary.wiley.com/doi/abs/10.1002/2014JD022365>

Please see our response for the last major comment above.

Reviewer #3 (Remarks to the Author):

No comments received.

References

1. Bollasina, Massimo A., Yi Ming, and V. Ramaswamy. "Anthropogenic aerosols and the weakening of the South Asian summer monsoon." *science* 334.6055 (2011): 502-505.
2. Vecchi, Gabriel A., et al. "Weakening of tropical Pacific atmospheric circulation due to anthropogenic forcing." *Nature* 441.7089 (2006): 73-76.
3. Cai, Wenju, et al. "Increasing frequency of extreme El Niño events due to greenhouse warming." *Nature climate change* 4.2 (2014): 111-116.
4. Andrews, Timothy, et al. "Accounting for changing temperature patterns increases historical estimates of climate sensitivity." *Geophysical Research Letters* 45.16 (2018): 8490-8499.

REVIEWER COMMENTS

Reviewer #2 (Remarks to the Author):

Review of Anthropogenic forcing yields significant positive trend in Earth's energy imbalance
by Shiv Priyam Raghuraman, David Paynter , V. Ramaswamy

Thank you for the revision. I find the additions and clarifications of the authors to help. In my view though, uncertainty in trends in the observations still must be dealt with more formally to achieve a publishable manuscript. My main comments:

1) My apologies for mistaking the Boucher and Held references - I actually thought the authors intended to cite Held et al. since it seemed to be the more relevant reference. I do suggest that the citation to Boucher et al. be clarified to include the figure # given how long the manuscript is and how deep into it the relevant figures are located.

2) Figure 2 caption: I suggest adding a bit more detail saying how the "maximum" is estimated, e.g. from how many years of data? From how many models?

3) Figures 2/3: I continue to assert that the CERES TEEI values not be shown as a single value but a PDF that incorporates uncertainty including drift uncertainty. From what I can tell this has not been done, including in Fig. 4. The authors say they have addressed this concern in the prior revision but I find their comments and approach to be deficient yet this is a central aspect of the current work. To assume instrument drift has no monthly autocorrelation defies logic. Moreover to assert that it has been entirely removed by the CERES team reflects a basic misunderstanding of the CERES data. In fact there is an unknown and arguably unknowable drift contained in the CERES estimates. Efforts have been made to account for it yet since "truth" is unknown the drift is difficult to remove since Terra and Aqua could be drifting in the same direction (and given the similarity of their hardware, likely are). In my personal communications with Norm Loeb, he estimates this to be

"0.2-0.3 Wm⁻² per decade". Clearly this is a zeroth order concern and I again implore the authors to deal with it correctly.

Line By Line Comments

45: how is the statement on OHC uptake acceleration justified? Just because 90% of EEI goes into the ocean on long time scales does not mean that change in rate also has to on short time scales?

Reviewer #2 (Remarks to the Author):

Review of Anthropogenic forcing yields significant positive trend in Earth's energy imbalance by Shiv Priyam Raghuraman, David Paynter , V. Ramaswamy

Thank you for the revision. I find the additions and clarifications of the authors to help. In my view though, uncertainty in trends in the observations still must be dealt with more formally to achieve a publishable manuscript.

We thank the reviewer for taking the time and effort to review our manuscript. Your suggestions improved the manuscript, and we thank you for that. We have now looked into and dealt in more depth with the observational uncertainty. Please see our responses below.

My main comments:

1) My apologies for mistaking the Boucher and Held references - I actually thought the authors intended to cite Held et al. since it seemed to be the more relevant reference. I do suggest that the citation to Boucher et al. be clarified to include the figure # given how long the manuscript is and how deep into it the relevant figures are located.

Thank you for the suggestion. We have now added the Figure numbers of the Boucher citation.

2) Figure 2 caption: I suggest adding a bit more detail saying how the “maximum” is estimated, e.g. from how many years of data? From how many models?

Thank you for the suggestion. We have now added more detail to the caption of Figure 2.

3) Figures 2/3: I continue to assert that the CERES TEEI values not be shown as a single value but a PDF that incorporates uncertainty including drift uncertainty. From what I can tell this has not been done, including in Fig. 4. The authors say they have addressed this concern in the prior revision but I find their comments and approach to be deficient yet this is a central aspect of the current work. To assume instrument drift has no monthly autocorrelation defies logic. Moreover to assert that it has been entirely removed by the CERES team reflects a basic misunderstanding of the CERES data. In fact there is an unknown and arguably unknowable drift contained in the CERES estimates. Efforts have been made to account for it yet since “truth” is unknown the drift is difficult to remove since Terra and Aqua could be drifting in the same direction (and given the similarity of their hardware, likely are). In my personal communications with Norm Loeb, he estimates this to be

"0.2-0.3 Wm⁻² per decade". Clearly this is a zeroth order concern and I again implore the authors to deal with it correctly.

Thank you for this suggestion and pointing us toward a particular value. We apologize for not adding an estimate of observational trend uncertainty (drift, instrument error, etc.) earlier since past literature did not indicate this was a large source of error (presumably because of the reason you mentioned: this being considered an ‘unknown unknown’). We have now updated all our Figures and text with an inclusion of 0.2 Wm⁻²decade⁻¹ observational uncertainty.

We contacted Norman Loeb to better understand the origins of the quoted 0.2-0.3 Wm⁻²decade⁻¹ uncertainty. During the 02/2012-07/2019 period (S-NPP Crosstrack period), Terra-Aqua difference in EEI trend is 0.23 Wm⁻²decade⁻¹. But over the much longer period of the Aqua satellite, Terra-Aqua = 0.05 Wm⁻²decade⁻¹. Furthermore, Norman Loeb indicated that CERES-Argo = 0.07 Wm⁻²decade⁻¹ (independent estimates, apart from a one-time mean EEI

adjustment in CERES using Argo). In light of this and keeping in mind the unknown nature of the drift, we assumed a $0.2 \text{ Wm}^{-2}\text{decade}^{-1}$. We describe below how our conclusions remain intact even in the presence of the larger uncertainty.

Using the broader model estimate of internal variability uncertainty, i.e., $\sigma_{Var.} = \frac{\epsilon}{2}$, in the revised Figure 4, yields the 95% CI ($1.96 * \sqrt{\sigma_{Obs.}^2 + \sigma_{Var.}^2}$; added in quadrature because internal variability uncertainty is independent from observational uncertainty) of CERES TEEI as $0.38 \pm 0.28 \text{ Wm}^{-2}\text{decade}^{-1}$. Furthermore, the 99% CI is $0.38 \pm 0.36 \text{ Wm}^{-2}\text{decade}^{-1}$. First, this shows that CERES TEEI > 0 and therefore continues to violate the null hypothesis that internal variability and/or drift error caused the observed TEEI. Second, this implies that there is < 0.5% probability (normal PDF) the mean CERES TEEI is less than $0.02 \text{ Wm}^{-2}\text{decade}^{-1}$, or, in other words, even with the large observational uncertainty, we are 99.5% confident that the anthropogenic forcing and response contribution has a positive value.

Similarly, in the revised Figure 2, although there is now a small overlap between the CERES range and the $\max(\epsilon)$ bar at $0.18\text{-}0.20 \text{ Wm}^{-2}\text{decade}^{-1}$, the probability that CERES has a low trend due to drift and that there was a high trend due internal variability in the same two-decade period is extremely small (< 1% probability). This is because the right-end tail of the internal variability probability distribution is being multiplied by the left-end tail of an assumed normal distribution of the observational uncertainty. Moreover, this result is insensitive to the probability density function (normal distribution, left-skew normal distribution, or uniform distribution) that is assumed for the observational uncertainty. We now describe this in the Methods too.

Line By Line Comments

45: how is the statement on OHC uptake acceleration justified? Just because 90% of EEI goes into the ocean on long time scales does not mean that change in rate also has to on short time scales? Thank you for pointing this out. We have deleted this sentence as it did not contribute anything extra to the existing content in the Introduction.

REVIEWERS' COMMENTS

Reviewer #2 (Remarks to the Author):

The authors have successfully addressed my concerns.

Reviewer #2 (Remarks to the Author):

The authors have successfully addressed my concerns.
We thank the reviewer for their comments.